# Characterizing superspreading potential of infectious disease: Decomposition of individual transmissibility

**Shi Zhao**[1,2]*, **Marc K. C. Chong**[1,2], **Sukhyun Ryu**[3], **Zihao Guo**[1], **Mu He**[4], **Boqiang Chen**[5], **Salihu S. Musa**[5,6], **Jingxuan Wang**[1], **Yushan Wu**[1], **Daihai He**[5]*, **Maggie H. Wang**[1,2]

**1** JC School of Public Health and Primary Care, Chinese University of Hong Kong, Hong Kong, China, **2** CUHK Shenzhen Research Institute, Shenzhen, China, **3** Department of Preventive Medicine, Konyang University College of Medicine, Daejeon, South Korea, **4** Department of Foundational Mathematics, Xi'an Jiaotong-Liverpool University, Suzhou, China, **5** Department of Applied Mathematics, Hong Kong Polytechnic University, Hong Kong, China, **6** Department of Mathematics, Kano University of Science and Technology, Wudil, Nigeria

* zhaoshi.cmsa@gmail.com (SZ); daihai.he@polyu.edu.hk (DH)

**Data Availability Statement:** All data used in this work were publicly available in literature, which were originally collected via the public domains. The processed data and codes are shared via

## Abstract

In the context of infectious disease transmission, high heterogeneity in individual infectiousness indicates that a few index cases can generate large numbers of secondary cases, a phenomenon commonly known as superspreading. The potential of disease superspreading can be characterized by describing the distribution of secondary cases (of each seed case) as a negative binomial (NB) distribution with the dispersion parameter, $k$. Based on the feature of NB distribution, there must be a proportion of individuals with individual reproduction number of almost 0, which appears restricted and unrealistic. To overcome this limitation, we generalized the compound structure of a Poisson rate and included an additional parameter, and divided the reproduction number into independent and additive fixed and variable components. Then, the secondary cases followed a Delaporte distribution. We demonstrated that the Delaporte distribution was important for understanding the characteristics of disease transmission, which generated new insights distinct from the NB model. By using real-world dataset, the Delaporte distribution provides improvements in describing the distributions of COVID-19 and SARS cases compared to the NB distribution. The model selection yielded increasing statistical power with larger sample sizes as well as conservative type I error in detecting the improvement in fitting with the likelihood ratio (LR) test. Numerical simulation revealed that the control strategy-making process may benefit from monitoring the transmission characteristics under the Delaporte framework. Our findings highlighted that for the COVID-19 pandemic, population-wide interventions may control disease transmission on a general scale before recommending the high-risk-specific control strategies.

https://github.com/plxzpnxZBD/Superspreading_withDelaporteDist.

**Funding:** DH was supported by Collaborative Research Fund [C7123-20G] of the Research Grants Council (RGC) of Hong Kong, China. MHW was supported by the National Natural Science Foundation of China [31871340, 71974165], Health and Medical Research Fund, the Food and Health Bureau, the Government of the Hong Kong Special Administrative Region [COVID190103, INF-CUHK-1], and the Chinese University of Hong Kong Grant [PIEF/Ph2/COVID/06, 4054600]. The funders had no role in study design, data collection and analysis, decision to publish, or preparation of the manuscript.

**Competing interests:** We have read the journal's policy and the authors of this manuscript have the following competing interests: MHW is a shareholder of Beth Bioinformatics Co., Ltd. Other authors declared no competing interests. The funding agencies had no role in the design and conduct of the study; collection, management, analysis, and interpretation of the data; preparation, review, or approval of the manuscript; or decision to submit the manuscript for publication.

## Author summary

Superspreading is one of the key transmission features of many infectious diseases and is considered a consequence of the heterogeneity in infectiousness of individual cases. To characterize the superspreading potential, we divided individual infectiousness into two independent and additive components, including a fixed baseline and a variable part. Such decomposition produced an improvement in the fit of the model explaining the distribution of real-world datasets of COVID-19 and SARS that can be captured by the classic statistical tests. Disease control strategies may be developed by monitoring the characteristics of superspreading. For the COVID-19 pandemic, population-wide interventions are suggested first to limit the transmission at a scale of general population, and then high-risk-specific control strategies are recommended subsequently to lower the risk of superspreading.

This is a *PLOS Computational Biology* Methods paper.

## 1 Introduction

The response to infectious disease epidemics can be improved by understanding the characteristics defining the potential to transmit infections between individuals [1]. An intriguing aspect of infectious disease transmission is the circumstances under which the etiological agent is transmitted to a large number of secondary cases from merely a proportion of primary cases [2–6]. The number of secondary transmissions per index case shows levels of heterogeneity [7], while overdispersion refers to transmission with high heterogeneity [8]. Such situations are considered consequences of heterogeneity in individual infectiousness and stochasticity in disease transmission [9, 10] as documented by numerous superspreading events [3, 11–16]. For example, superspreading potentials and traceable events of COVID-19 transmission have frequently been reported in terms of a scale of $k$ estimates [12, 17–19], which appear similar to those of previous epidemics of SARS and Middle East respiratory syndrome coronavirus (MERS-CoV) [5, 20–22]. The heterogeneity in transmission is determined by many factors including the characteristics of the host and the pathogen [23], the mode and setting of transmission [17, 24], the contact patterns [25], the viability of the pathogen, and the environmental components [8, 26–28]. Risk management and disease control strategies may vary and may be adjusted in response to different levels of individual heterogeneity in transmission [11, 29–31]. Thus, methods used to characterize heterogeneity in transmission are a public health priority to better understand patterns in infectious disease transmission [32] and in specifying informed control strategies [29, 33–36].

On one hand, the reproduction number $R$ is commonly adopted to measure the average (or expected) number of secondary cases generated by a typical infectious individual [37]. The scales of $R$ were sometimes given unwarranted priority in the assessment of pandemic potential [2, 38, 39], which means that $R$ cannot reflect the scale of heterogeneity in individual infectiousness [40–43]. On the other hand, by acknowledging the heterogeneity in disease transmission patterns, a negative binomial (NB) distribution has been widely applied as a model for count data [44], particularly for offspring cases data that exhibit overdispersion [29], that is, with variance that is greater than the mean values. As such, the heterogeneity in

transmission can be quantified by describing the distribution of secondary cases generated by each index case as the NB distribution with dispersion parameter, $k$ [44]. The conceptualization of a NB distribution incorporates the stochastic effects of disease transmission [9] and the variability in individual infectiousness [29]. Mathematically, the framework for the NB distribution was formulated by compounding a Poisson distribution with a Gamma-distributed rate parameter, where the dispersion parameter $k$ accounts for the variation in individual infectiousness reflected in the Gamma distribution [45]. This NB framework was widely adopted and yielded better fitting performance (against the Poisson distribution) in governing real-world observations of offspring cases or cluster sizes [17, 29, 40, 46]. A smaller $k$ value suggests that transmission is more dispersive, and therefore outbreaks are likely to involve superspreading events [3]. When $R$ is fixed, a smaller $k$ corresponds to a lower effectiveness of non-pharmaceutical interventions in controlling epidemics [30, 47].

Regarding the description of heterogeneity in transmission from a theoretical standpoint, candidate models have been compared based on their fitting performances to real-world observations [29]. Inspired by the compounding relationship between Poisson and NB distributions, we considered that the composition of the Poisson rate can be modelled using a more generalized framework. In this study, to explain the heterogeneity in the distribution of offspring, we propose the application of the Delaporte distribution, which is a generalized version of the NB distribution and can also be derived by compounding the Poisson rate [48, 49]. By fitting several datasets of offspring (or secondary) cases, we illustrated that the Delaporte distribution led to an improved or equivalent fitting performance compared to the NB distribution, and this improvement becomes more evident as the sample size increases. For model selection using the likelihood ratio (LR) test, the Delaporte distribution demonstrated increasing statistical power but a conservative type I error rate for a wide range of sample sizes. We highlight the potential of the Delaporte distribution in quantifying the superspreading characteristics of infectious diseases and for recommending disease control strategies.

## 2 Methods

### 2.1 Decomposition of the variation in individual infectiousness

Following the classic theoretical framework of disease transmission [9], stochastic effects in transmission are considered to have a Poisson distribution [50], which is denoted $X \sim \text{Poisson}(\lambda)$. Here, the random variable $X$ denotes the number of secondary cases caused by a randomly-selected primary case, and the parameter $\lambda$ is the Poisson rate. To account for the variation in individual infectiousness, the Poisson rate $\lambda$ is a variable attribute among different hosts, and thus the distribution of $X$ becomes a Poisson mixture, as proposed previously in [29].

We then decomposed the offspring number ($X$) of each index case into two components, including a fixed part ($X_F$) and variable part ($X_V$), such that $X_F + X_V = X$. Here, $X_F$ and $X_V$ were assumed to be independent variables and followed the compound Poisson distributions with rate parameters ($\lambda_F$ and $\lambda_V$) that followed two Gamma distributions, so that $\lambda_F \sim \text{Gamma}$ (mean = $R_F$, dispersion = $k_F$), and $\lambda_V \sim \text{Gamma}$(mean = $R_V$, dispersion = $k_V$). This was equivalent to the Poisson rate $\lambda$ that was directly decomposed into two independent additive components denoted by $\lambda = \lambda_F + \lambda_V$ [49], where both $\lambda_F$ and $\lambda_V$ are nonnegative values. As such, $X$ is the sum of two independent negative-binomial distributed variables. Referring to the definition in [29], $\lambda$ was conceptualized as the individual reproduction number [51], which is a random variable and represents the expected number of secondary cases caused by a (particularly) given primary case.

For the fixed component ($X_F$), we modelled $k_F \to \infty$ assuming there was no variation in the fixed part ($\lambda_F$) of individual infectiousness. By denoting the probability mass function (PMF) of $X$ as $f_D(X)$, the probability of generating function (PGF), $g_D(\cdot)$, was as follows:

$$g_D(s) = \mathbf{E}[s^{X_F}] \cdot \mathbf{E}[s^{X_V}] = \lim_{k_F \to \infty} \left[1 + \frac{R_F}{k_F}(1-s)\right]^{-k_F} \cdot \left[1 + \frac{R_V}{k_V}(1-s)\right]^{-k_V}$$

$$= \exp[-R_F(1-s)] \cdot \left[1 + \frac{R_V}{k_V}(1-s)\right]^{-k_V}$$

Because the term $k_F$ vanishes, we denoted $k_V$ by $k$ for convenience. The $\lambda_F$ is the fixed component, which is a constant, and we defined $R_F = \lambda_F$. The $\lambda_V$ is the variable component, which follows a Gamma distribution with a mean $R_V$ and dispersion (or shape) parameter $k$. Mathematically, $X \sim \text{Poisson}(\lambda_F + \lambda_V)$ on the condition that $\lambda_V \sim \text{Gamma}(\text{mean} = R_V, \text{dispersion} = k)$. Then, the PGF $g_D(\cdot)$ is defined as shown in Eq (1):

$$g_D(s) = \exp[-R_F(1-s)] \cdot \left[1 + \frac{R_V}{k}(1-s)\right]^{-k} \tag{1}$$

By identifying the PGF $g_D(\cdot)$, we find that the distribution of $X$ was a Delaporte distribution, denoted by $f_D(\cdot)$, with parameters $R_F$, $R_V$, and $k$.

If we define $R = R_F + R_V$, $R$ is the population reproduction number as the expected (or average) number of secondary cases caused by a (typical) primary case [52, 53], and thus we have $R = \mathbf{E}[X] = \mathbf{E}[\lambda]$, where $\mathbf{E}[\cdot]$ is the expectation function. The $R_F$ and $R_V$ account for the fixed and variable components of the reproduction number ($R$), and thus we have $R = \mathbf{E}[X] = \mathbf{E}[X_F] + \mathbf{E}[X_V] = \mathbf{E}[\lambda] = \mathbf{E}[\lambda_F] + \mathbf{E}[\lambda_V] = R_F + R_V$, which is the mean of the Delaporte distribution $f_D(X)$. As such, $X_F$ and $X_V$ are components of the (observable) number of offspring cases $X$, $\lambda_F$ and $\lambda_V$ are components of the (latent) individual reproduction number $\lambda$, which is a variable, and $R_F$ and $R_V$ are components of the population reproduction number $R$, which is considered as a constant. In particular, the distribution function of $\lambda$ has both a discrete part and a continuous part.

**2.1.1 Delaporte distribution.** Under the formulation of a Delaporte distribution [48], the probability mass function (PMF) $f_D(X)$ has three parameters, $R_F$, $R_V$, and $k$, and is given in Eq (2).

$$f_D(X = x) = \sum_{a=0}^{x} \left[ \frac{\Gamma(k+a)}{\Gamma(k)\Gamma(a+1)} \left(\frac{k}{R_V + k}\right)^k \left(\frac{R_V}{R_V + k}\right)^a \cdot \frac{R_F^{x-a} \cdot \exp(-R_F)}{\Gamma(x-a+1)} \right]$$

$$= \sum_{a=0}^{x} \frac{\Gamma(k+a) \cdot \left(\frac{R_V}{k}\right)^a \cdot R_F^{x-a} \cdot \exp(-R_F)}{\Gamma(k)\Gamma(a+1) \cdot \left(1 + \frac{R_V}{k}\right)^{k+a} \cdot \Gamma(x-a+1)} \tag{2}$$

Here, $\Gamma(\cdot)$ denotes the Gamma function, and the integer $x$ denotes the number of secondary cases. Eq (2) can be considered as a 'convolution' between an NB distribution and a Poisson distribution.

Compared to the classic NB distribution proposed in [29], the Delaporte distribution can be restricted to an NB distribution if $R_F = 0$, or equivalently $R_V = R$. Similarly, if $R_V = 0$ or $k \to \infty$, the Delaporte distribution is restricted to a Poisson distribution [49]. Thus, either the NB or Poisson distribution is a special case of Delaporte distribution. Let the fraction of the fixed component $\rho$ be defined as $\rho = R_F / R$, and straightforwardly, we have $0 \le \rho \le 1$. Equivalently, $f_D(X)$ in Eq (2) can also be formulated in an alternative version by replacing $R_F$ with $\rho R$ and $R_V$

with $(1-\rho)R$, which is expressed in Eq (3),

$$f_{\mathrm{D}}(X = x) = \sum_{a=0}^{x} \left[ \frac{\Gamma(k+a)}{\Gamma(k)\Gamma(a+1)} \left( \frac{k}{(1-\rho)R+k} \right)^{k} \left( \frac{(1-\rho)R}{(1-\rho)R+k} \right)^{a} \cdot \frac{(\rho R)^{x-a} \cdot \exp(-\rho R)}{\Gamma(x-a+1)} \right] =$$

$$\sum_{a=0}^{x} \frac{\Gamma(k+a) \cdot \left( \frac{(1-\rho)R}{k} \right)^{a} \cdot (\rho R)^{x-a} \cdot \exp(-\rho R)}{\Gamma(k)\Gamma(a+1) \cdot \left( 1 + \frac{(1-\rho)R}{k} \right)^{k+a} \cdot \Gamma(x-a+1)}$$

(3)

Here, the three parameters for the Delaporte distribution change to $R$, $\rho$, and $k$. As such, the Delaporte distribution becomes a Poisson distribution when $\rho = 1$, or a NB distribution when $\rho = 0$, that is, $f_{\mathrm{NB}}(x) = \frac{\Gamma(k+x)}{\Gamma(k)\Gamma(x+1)} \left( \frac{k}{R+k} \right)^{k} \left( \frac{R}{R+k} \right)^{x}$.

The variance of $X$ is derived as $\mathbf{Var}(X) = \rho R + (1-\rho)R\left(1 + \frac{(1-\rho)R}{k}\right)$ under the formula in Eq (3), or $\mathbf{Var}(X) = R_{\mathrm{F}} + R_{\mathrm{V}}\left(1 + \frac{R_{\mathrm{V}}}{k}\right)$ using the formula in Eq (2) in alternative. We derive $\frac{\mathrm{d}\mathbf{Var}(X)}{\mathrm{d}\rho} \leq 0$ for $0 \leq \rho \leq 1$, and $\frac{\mathrm{d}\mathbf{Var}(X)}{\mathrm{d}k} < 0$. Because $\mathbf{Var}[X]$ reflects the scale of variation in individual infectiousness, a smaller value for either $\rho$ or $k$ indicates a higher level of transmission heterogeneity or superspreading potential.

The implementation of Delaporte distribution is considered a generalization of the framework proposed in [29], and thus the interpretation of the dispersion parameter $k$ generalizes its meaning in the NB distribution [45]. As the fixed part ($R_{\mathrm{F}}$) of $R$ vanishes in the NB distribution, $1/\sqrt{k}$ is the coefficient of variation (CV) of the Gamma distribution followed by the individual reproduction numbers ($\lambda$). In the context of the Delaporte distribution, the effect of $k$ on shaping the variation of $\lambda$ is restricted to the CV of its variable part ($\lambda_{\mathrm{V}}$), which is also Gamma-distributed.

Differences in the PMF of Poisson, NB, and Delaporte distributions are shown in Fig 1.

**2.1.2 Epidemiological measurements of heterogeneity in transmission.** In epidemiological studies [3, 4, 17, 54], the heterogeneity in disease transmission is frequently reported as a general '20/80' rule [21, 55], that is, according to the Pareto principle, whereby 20% of primary cases cause 80% of secondary cases [56]. With the three parameters of the Delaporte distribution, the transmission distribution profiles can be translated in the form of the '20/80' rule. Following the framework proposed in [3, 57], the proportion ($0 \leq Q \leq 1$) of secondary cases can

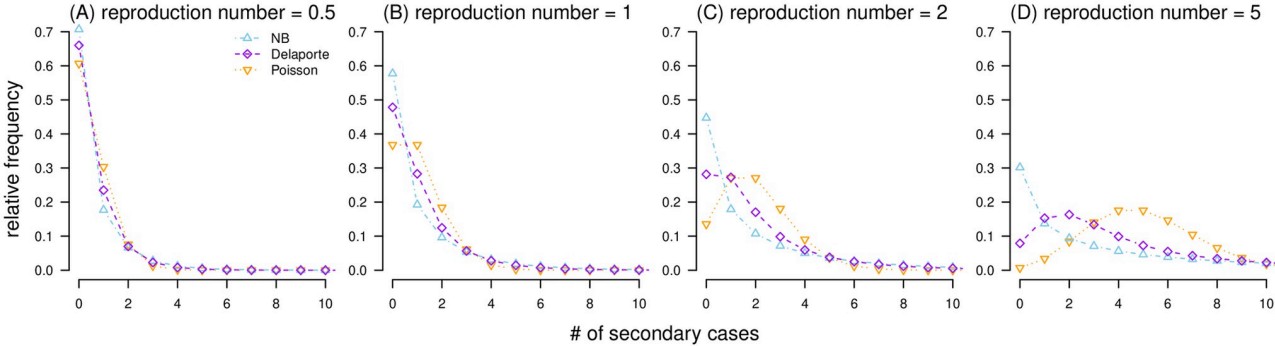

**Fig 1. Probability mass functions (PMF) of Poisson (in orange), negative binomial (in blue), and Delaporte (in purple) distributions.** In each panel, the dispersion parameter $k$ is fixed at 0.5, and the fraction of fixed component $\rho$ is fixed at 0.3.

be determined by the transmission contributed by a proportion ($0 \leq P \leq 1$) of the most infectious primary cases [33], and *vice versa*, which was formulated in Eq (4).

$1 - P = \int_0^Z f_D(X = \lfloor z \rfloor) dz$, and the variable $Z$ satisfies

$$1 - Q = \frac{\int_0^Z \lfloor z \rfloor \cdot f_D(X = \lfloor z \rfloor) dz}{\sum_{x=0}^{\infty} z \cdot f_D(X = x)} \tag{4}$$

Here, $\lfloor \cdot \rfloor$ denotes the floor function, which outputs the largest integer less than or equal to the given number. Note that the $\sum_{x=0}^{\infty} z \cdot f_D(X = x)$ at the denominator is the mean of the Delaporte distribution, i.e., $R_F + R_V$ or $R$. Conventionally, $Q$ is fixed at 0.8, and the value of $P$ is of interest. A smaller $P$ indicates that a smaller but core proportion of high-risk cases may generate most offspring cases, indicating a higher level of heterogeneity in transmission.

Generally, $Q$ is considered a function of $P$, which is bound between 0 and 1 for both $Q$ and $P$. The concaveness of this '$Q$-$P$' function is positively related to the level of transmission heterogeneity [29], which is constructed in the same manner as the Lorenz curve [58, 59]. For a perfectly homogeneous scenario, where $X = R$ is a constant, we have $Q = P$.

Another important measurement of transmission heterogeneity is the proportion of primary cases that generate 0 secondary cases, which is given as $f_D(0) = f_D(X = 0)$ based on Eqs (2) or (3). With the reproduction number $R$ fixed, a larger value of $f_D(0)$ implies a higher level of heterogeneity in transmission.

## 2.2 Datasets

We adopted six sets of contact tracing data and extracted the observations of offspring cases generated by each seed case for further exemplification. These included five COVID-19 datasets collected in mainland China (dataset #1), South Korea (datasets #2a and b), Hong Kong (dataset #3), and Tianjin, China (dataset #4), and one SARS dataset collected in Beijing, China (dataset #5). The transmission chains within each dataset were screened and then reconstructed with systematic and strict 'inclusion-and-exclusion' screening criteria based on plausible epidemiological evidence and rigorous consistency checks. All datasets were previously published and adopted for analysis in peer-reviewed studies.

**2.2.1 Dataset #1: COVID-19 data in mainland China.**   For dataset #1, we used the COVID-19 contact tracing data published in [12], which was accessed freely via the public repository https://github.com/linwangidd/covid19_transmissionPairs_China/blob/master/transmission_pairs_covid_v2.csv. The same dataset was also adopted to estimate the dispersion parameter in [30].

Dataset #1 contains 1407 transmission pairs that were identified and reconstructed in previous studies, governmental news release, and official situation reports from 15 January to 29 February 2020 in mainland China. We identified 807 infectors with at least one secondary case and extracted the number of offspring infectees generated by each infector. A total of 1241 sporadic or terminal cases with 0 secondary cases were identified. Thus, dataset #1 includes observations of secondary case numbers with a sample size of 2048.

**2.2.2 Datasets #2a and #2b: COVID-19 data in South Korea.**   For datasets #2a and #2b, we used the COVID-19 contact tracing data published in [33], which were shared by the authors. Both datasets shared the same source of information from the local public health authorities in South Korea, excluding the Daegu-Gyeongsangbuk region, where the data were not publicly reported.

Referring to [33], the original dataset was divided into different periods according to the onset dates of infectors. Dataset #2a contains 571 infectors with at least one secondary case and 830 sporadic or terminal cases during the epidemic period from 20 April to 16 October 2020. Dataset #2b contains 104 infectors and 240 sporadic or terminal cases occurring during the epidemic period from 19 January to 19 April 2020. As such, datasets #2a and #2b include observations of secondary case numbers with sample sizes of 1401 and 344, respectively.

**2.2.3 Dataset #3: COVID-19 data in Hong Kong.** For dataset #3, we used the COVID-19 contact tracing data published in [17], which was accessed freely via public repository, https://github.com/dcadam/covid-19-sse/blob/master/data/transmission_pairs.csv. Dataset #3 contains 169 transmission pairs that were identified and reconstructed according to governmental news releases and official situation reports published on 7 May 2020 in Hong Kong [60, 61]. There were 91 infectors, 153 terminal cases, and 46 local sporadic cases identified, and we extracted information on the number of offspring infectees generated by each infector. As such, dataset #3 included observations of secondary case numbers with a sample size of 290 cases.

**2.2.4 Dataset #4: COVID-19 data in Tianjin, China.** For dataset #4, we used the COVID-19 contact tracing data published in [19], which was freely obtained from the supplementary materials, accessed via https://www.mdpi.com/1660-4601/17/10/3705/s1. Dataset #4 contained 36 clusters of cases, including 47 cases of COVID-19, which were identified and reconstructed according to a governmental news releases and official situation reports between 21 January and 26 February 2020 in Tianjin, China [62], and each cluster was caused by a primary case. We identified seven infectors with 11 associated terminal cases and 29 local sporadic cases. Thus, dataset #4 contains observations of secondary case numbers with a sample size of 47.

**2.2.5 Dataset #5: SARS data in Beijing, China.** For dataset #5, we used the SARS contact tracing data of superspreading events from April to May 2003 previously published in [5], which was also attempted to estimate the dispersion parameter in [29]. The 34 cases in the first and second generation were considered the source cases, and we extracted the number of offspring infectees generated by each source case. Thus, dataset #5 contained observations of secondary case numbers with a sample size of 34.

## 2.3 Likelihood framework and statistical inference

We considered the number of secondary cases observed from each primary case with a sample size $N$. Considering the infector who generates $j$ ($\geq 0$) secondary cases, or equivalently a cluster of cases with size $(j + 1)$ in one transmission generation, we denoted the number of these infectors by $n_j$. Then, similar to previous studies [3, 17], the likelihood of observing $n_j$ clusters with size $(j + 1)$ was $[f_D(X = j)]^{n_j}$. Thus, we construct the overall log-likelihood function, $\ell$, in Eq (5).

$$\ell = \log(L) = \log\left[\prod_{j \geq 0}[f_D(X = j)]^{n_j}\right] \tag{5}$$

Hence, $\sum_{j \geq 0} n_j = N$.

To match the real-world observations, we adopted a Bayesian fitting procedure with a Metropolis–Hastings Markov chain Monte Carlo (MCMC) algorithm with non-informative prior distributions for parameter estimation. Based on the likelihood in Eq (5), the MCMC was conducted with five chains and 100,000 iterations for each chain, including 40,000 iterations for the burn-in period, to obtain the posterior estimates. The convergence of each MCMC chain was visually checked using trace plots and the Gelman–Rubin–Brooks diagnostic quantitatively [63]. The median and 95% credible intervals (95%CrI) of the posterior

distributions of $R_F$, $R_V$, and $k$ were calculated and summarized for comparison with the previous estimates and across each dataset.

For comparisons with the classic Poisson or NB framework, we also repeated the estimation procedures by restricting $R_F = R$ (i.e., $R_V = 0$) for the Poisson distribution, or $R_F = 0$ (i.e., $R_V = R$) for the NB distribution.

## 2.4 Evaluation of fitting and testing performance

In accordance with previous study [17], the Akaike information criterion (AIC) of MLE was used to measure the fitting performance of the Poisson, NB, and Delaporte distributions. Statistical evidence supporting the improvement in the fitting performance is claimed when the AIC units are reduced by 2 or more [40, 64].

The likelihood ratio (LR) test was adopted to assess the statistical significance of the improvement (in goodness-of-fit) of the Delaporte distribution versus the NB distribution. The test statistic ($\pi^*$) of the LR test was given as follows.

$$\pi^* = 2 \cdot [\log(L) - \log(L_{NB})] \sim \text{Chi}(\text{df} = 1)$$

where $L_{NB}$ denotes the likelihood of the NB distribution and $L$ denotes the likelihood of the Delaporte distribution. Therefore, the $p$-value was calculated as the percentile of the Chi-squared distribution with degree of freedom df = 1 [11], which was expressed as follows:

$$p\text{-value} = \text{pChi}(\pi^*|\text{df} = 1).$$

Here, pChi ($\cdot$) denotes 1 minus the cumulative distribution function (i.e., survival function) of the Chi-squared distribution. Similar frameworks have also been adopted in previous studies [46, 64–66]. We considered $p$-value $< 0.05$ as a statistically significant improvement of the Delaporte distribution compared to the NB distribution, and thus the Delaporte distribution was selected as an optimization. Note that this appears statistically equivalent to having a significant estimate of $0 < \rho < 1$, or both $R_F$ and $R_V > 0$.

To test performance, the power and type I error of the LR test were evaluated. The testing power is calculated as the probability of $p$-value $< 0.05$ for fitting Delaporte distribution to the real-world observations compared to the NB distribution. We generated pseudo-datasets with different sample sizes by random sampling with replacement, a method similar to non-parametric bootstrapping, from the datasets described in Section 2.2. The type I error rate was calculated as the probability of $p$-value $< 0.05$ for fitting Delaporte distribution to the NB distributed datasets against the NB distribution. We generated the NB-distributed datasets with Monte Carlo random sampling from NB distributions. Note that statistically, the $p$-value $< 0.05$ from the LR test here was (roughly) equivalent to the AIC-based model selection with a cutoff of 2 units.

The parameter estimation of NB, and Delaporte distribution was obtained for each pseudo- or NB-distributed dataset using the approach described in Section 2.3. We summarized the test statistic ($\pi^*$), power, and type I error rate based on the different sample sizes.

## 2.5 Extension of other types of real-world observations

Although helpful in estimating superspreading potentials, the number of offspring cases per index case in our dataset section was not always accurately reported [46]. In many situations, it is time or financially consuming for surveillance procedures to collect these datasets [67], and it is also difficult to maintain the consistency of reporting standards or secure sufficient samples [68]. Alternatively, the cluster size of next transmission generation, i.e., the one-generation cluster size, and the final outbreak size including a few seed cases are also commonly adopted

to inform the characteristics of transmission. Thus, the theoretical frameworks in the following two sections were formulated to associate both types of real-world observations with the Delaporte distribution.

**2.5.1 Next-generation cluster size.** Cluster size data are frequently adopted to construct a statistical estimation [3, 40, 66]. Each one-generation cluster size observation is reported as the numbers of primary and secondary cases within a single transmission generation, which can also be simply translated into a number of primary cases and the cluster size of next-generation secondary cases. We discuss below the mathematical formulation of the distribution and likelihood function of a next-generation case cluster produced by a certain number of seed cases.

For a one-generation cluster of cases with size $(i + j)$, that is, within a single transmission generation, where $i\ (> 0)$ infectors generate $j\ (\geq 0)$ infectees, we consider the summation of $i$ independent and identically distributed (IID) random variables following the Delaporte distribution. Then, given the values of $R_F$, $R_V$, and $k$, the probability of observing an event in which $i$ $(\geq 0)$ infectors generate $j\ (\geq 0)$ infectees can be formulated by employing the probability generating function (PGF) $g_D(\cdot)$ in Eq (1). Thus, the PGF of the PMF of infectees number ($j$) generated by $i$ infectors, $h_D(\cdot)$, was as follows:

$$G(s) = [g_D(s)]^i = \exp[-(R_F i)(1 - s)] \cdot \left[1 + \frac{R_V i}{ki}(1 - s)\right]^{-ki}$$

By identifying the PGF $G(\cdot)$, we found that the distribution of the number infectees $j$ generated by $i$ infectors was also a Delaporte distribution, $h_D(j|i)$, with the parameters $R_F i$, $R_V i$, and $ki$, which was formulated as in Eq (6).

$$h_D(j|i) = \sum\nolimits_{a=0}^{j} \left[\frac{\Gamma(ki + a)}{\Gamma(ki)\Gamma(a + 1)}\left(\frac{k}{R_V + k}\right)^{ki}\left(\frac{R_V}{R_V + k}\right)^{a} \cdot \frac{(R_F i)^{j-a} \cdot \exp(-R_F i)}{\Gamma(j - a + 1)}\right] \qquad (6)$$

Alternatively, $h_D(\cdot)$ in Eq (6) could also be transformed by replacing $R_F i$ with $\rho R i$ and $R_V$ with $(1 - \rho)R$, which was expressed as follows,

$$h_D(j|i) = \sum_{a=0}^{j} \left[\frac{\Gamma(ki + a)}{\Gamma(ki)\Gamma(a + 1)}\left(\frac{k}{(1 - \rho)R + k}\right)^{ki}\left(\frac{(1 - \rho)R}{(1 - \rho)R + k}\right)^{a} \cdot \frac{(\rho R i)^{j-a} \cdot \exp(-\rho R i)}{\Gamma(j - a + 1)}\right]$$

It should be noted that for the new Delaporte distribution here, or in Eq (6), the fraction of fixed component ($\rho$) holds unchanged. As such, the likelihood function can be directly constructed by rearranging Eq (6) when one-generation cluster size observations were used to infer superspreading characteristics, that is, $\rho$ and $k$.

When $\rho$ approaches 0, the Delaporte distribution reduces to the NB distribution [49], and thus the 'convolution' in the equation above vanished, i.e., $a = j$. Then, the distribution of the number of infectees $j$ generated by $i$ infectors was from the NB distribution ($h_{NB}$),

$$h_{NB}(j|i) = \lim_{\rho \to 0^+} h_D(j|i) = \frac{\Gamma(ki + j)}{\Gamma(ki)\Gamma(j + 1)}\left(\frac{k}{R + k}\right)^{ki}\left(\frac{R}{R + k}\right)^{j}$$

which was also derived or adopted in previous studies [3, 4, 11, 17, 19, 22, 40, 46, 69]. Likewise, by using the branching process approach to characterize the size distribution introduced in [40, 69, 70], the formulation of Eq (6) can also be derived by obtaining the $j$-th derivative of

$g_\mathrm{D}(\cdot)$ at 0 according to the property of PGF [71], which means the following relationship holds.

$$\frac{1}{\Gamma(j+1)} \cdot \frac{\mathrm{d}^j [g_\mathrm{D}(s)]^i}{\mathrm{d}s^j}\Bigg|_{s=0} = h_\mathrm{D}(j|i)$$

which can be shown algebraically or by mathematical induction (details omitted).

**2.5.2 Final outbreak size with subcritical transmission.** Many outbreaks occur in the form of isolated cases, short chains of transmission, or small clusters [3, 72], for example, diseases with weak human-to-human transmission [68] or vaccine-preventable infections in a vaccine-available setting [73]. Thus, offspring cases observations like those in our data section are limited and difficult to access because the transmission is unlikely to be sustained. These outbreaks are recognized as subcritical (or self-limited) outbreaks when the population reproduction number appears to be less than 1 [11, 69], that is, $R < 1$, namely a weakly transmitting disease. Although the final outbreak size is frequently linked to subcritical transmission, the final outbreak size may also be observable for supercritical transmission ($R > 1$), which we will introduce below more rigorously. Each self-limited outbreak includes a group of cases connected by an unbroken series of transmission events (or chains), which was named the 'stuttering transmission chain' in [11].

Except for the first $i$ seed (or imported) cases, each case in a self-limited outbreak must be produced by one of the total cases with size denoted by $c$. According to [11], each secondary case must be linked to one of the other cases. Thus, the probability of observing a stuttering chain (or self-limited outbreak) size $c$ ($\geq i$) including $i$ ($> 0$) cases is ($i/c$) and multiplies the probability of $c$ primary cases causing ($c$–$i$) secondary cases in one generation, i.e., $\frac{i}{c} \cdot h_\mathrm{D}(c - i|c)$. In other words, under the independent and identically distributed assumption of the branching process [71], the probability of having a stuttering chain of size $c$ including $i$ cases, denoted by $\omega_\mathrm{D}(c, i)$, is the ($c - i$)-th coefficient of $\left[\frac{i}{c} \cdot [g_\mathrm{D}(s)]^c\right]$, which is equivalent to $\frac{i}{c} \cdot h_\mathrm{D}(c - i|c)$. Hence, we have

$$\omega_\mathrm{D}(c, i) = \frac{i}{c} \cdot \frac{1}{\Gamma(c - i + 1)} \cdot \frac{\mathrm{d}^{c-i}[g_\mathrm{D}(s)]^c}{\mathrm{d}s^{c-i}}\Bigg|_{s=0} = \frac{i}{c} \cdot h_\mathrm{D}(c - i|c)$$

The term $\frac{i}{c}$ is the normalization factor for the correction that $i$ out of $c$ cases are seed cases. This equation matches the relation derived in [40], which was also adopted in [57].

Rearranging the expression algebraically, we derive the exact formula of $\omega_\mathrm{D}(c, i)$ in Eq (7).

$$\omega_\mathrm{D}(c, i) = \frac{i}{c} \sum_{a=0}^{c-i} \left[ \frac{\Gamma(kc + a)}{\Gamma(kc)\Gamma(a + 1)} \left( \frac{k}{R_\mathrm{V} + k} \right)^{kc} \left( \frac{R_\mathrm{V}}{R_\mathrm{V} + k} \right)^a \cdot \frac{(R_\mathrm{F}c)^{c-i-a} \cdot \exp(-R_\mathrm{F}c)}{\Gamma(c - i - a + 1)} \right] \quad (7)$$

By replacing $R_\mathrm{F}i$ with $\rho Ri$ and $R_\mathrm{V}i$ with $(1 - \rho)Ri$, an alternative version of $\omega_\mathrm{D}(c, i)$ was expressed as follows,

$$\omega_\mathrm{D}(c, i) = \frac{i}{c} \sum_{a=0}^{c-i} \left[ \frac{\Gamma(kc + a)}{\Gamma(kc)\Gamma(a + 1)} \left( \frac{k}{(1 - \rho)R + k} \right)^{kc} \left( \frac{(1 - \rho)R}{(1 - \rho)R + k} \right)^a \cdot \frac{(\rho Rc)^{c-i-a} \cdot \exp(-\rho Rc)}{\Gamma(c - i - a + 1)} \right]$$

Therefore, the likelihood function can be constructed based on Eq (7) when stuttering chain size observations are available. When $\rho$ approaches 0, the Delaporte distribution reduces to the NB distribution [49], and thus $a = c - i$. Thus, the probability of observing the final

outbreak size $c$ including $i$ cases based on the NB distribution ($\omega_{\text{NB}}$),

$$\omega_{\text{NB}}(c, i) = \lim_{\rho \to 0^+} \omega_{\text{D}}(c, i) = \frac{i}{c} \cdot \frac{\Gamma(kc + c - i)}{\Gamma(kc)\Gamma(c - i + 1)} \left(\frac{k}{R + k}\right)^{kc} \left(\frac{R}{R + k}\right)^{c-i} = \frac{i}{c} \cdot h_{\text{NB}}(c - i | c)$$

Alternatively, the form below of $\omega_{\text{NB}}(c, i)$ was previously adopted, which was mathematically equivalent.

$$\omega_{\text{NB}}(c, i) = \frac{ki}{kc + (c - i)} \cdot \binom{kc + (c - i)}{c - i} \cdot \left(\frac{k}{R + k}\right)^{kc} \left(\frac{R}{R + k}\right)^{c-i}$$

Here, $\frac{i}{c} \cdot \frac{\Gamma(kc+c-i)}{\Gamma(kc)\Gamma(c-i+1)} = \frac{i}{c} \cdot \frac{kc}{kc+c-i} \cdot \frac{\Gamma(kc+c-i+1)}{\Gamma(kc+1)\Gamma(c-i+1)} = \frac{ki}{kc+(c-i)} \cdot \binom{kc+(c-i)}{c-i}$, and $\binom{kc+(c-i)}{c-i}$ is the combination function calculating number of elements' combinations with size $(c - i)$ can be selected from a population of elements with size $[kc + (c - i)]$. This formula was also adopted previously in [57].

As reported in [11, 69], with adjustment, the formula in Eq (7) is also applicable for supercritical transmission. When $R > 1$, there is a chance of $\left[1 - \sum_{c=i}^{\infty} \omega_{\text{D}}(c, i)\right]$ that the outbreak will never be extinct, which means the final outbreak size $c$ becomes a defective random variable. Based on the property of the branching process, we may calculate the probability of outbreak extinction $\varepsilon$ by solving $\varepsilon = [g_{\text{D}}(\varepsilon)^i]$ [69]. Thus, the likelihood function can also be constructed by adjusting $\varepsilon$ as the denominator for supercritical transmission.

Of particular interest is the final size of the outbreak generated by single seed case, i.e., $i = 1$, which is the probability of $c$ ($\geq 1$) primary cases causing $(c - 1)$ secondary cases, i.e., $h_{\text{D}}$ ($j = c - 1 | i = c) = h_{\text{D}}$ $(c - 1 | c)$, as in Eq (8).

$$\omega_{\text{D}}(c, 1) = \frac{1}{c} \sum_{a=0}^{c-1} \left[ \frac{\Gamma(kc + a)}{\Gamma(kc)\Gamma(a + 1)} \left(\frac{k}{R_{\text{V}} + k}\right)^{kc} \left(\frac{R_{\text{V}}}{R_{\text{V}} + k}\right)^{a} \cdot \frac{(R_{\text{F}}c)^{c-a-1} \cdot \exp(-R_{\text{F}}c)}{\Gamma(c - a)} \right] \quad (8)$$

which was translated by rearranging Eq (6) and can alternatively be expressed as follows,

$$\omega_{\text{D}}(c, 1) = \frac{1}{c} \sum_{a=0}^{c-1} \left[ \frac{\Gamma(kc + a)}{\Gamma(kc)\Gamma(a + 1)} \left(\frac{k}{(1 - \rho)R + k}\right)^{kc} \left(\frac{(1 - \rho)R}{(1 - \rho)R + k}\right)^{a} \cdot \frac{(\rho Rc)^{c-a-1} \cdot \exp(-\rho Rc)}{\Gamma(c - a)} \right]$$

When $\rho$ approaches 0, we have the NB version, $\omega_{\text{NB}}(c, 1)$, as follows,

$$\omega_{\text{NB}}(c, 1) = \frac{1}{c} \cdot \frac{\Gamma(kc + c - 1)}{\Gamma(kc)\Gamma(c)} \left(\frac{k}{R + k}\right)^{kc} \left(\frac{R}{R + k}\right)^{c-1}$$

which is consistent with the formula derived or used in previous studies [3, 11, 33, 40, 69]. Note that $c \cdot \Gamma(c) = \Gamma(c + 1)$.

## 2.6 Theoretical framework of different control schemes

We formulated the following two control schemes (**I**) and (**II**) with same reduction amount in reproduction number and compared their respective control efficacies in reducing the risks of superspreading [outcome (**I**)] or outbreak [outcome (**II**)]. For both schemes, we considered the control effect ($\xi$) in terms of the fractional reduction in the reproduction number ($R$), where $\xi = 0$ reflects no control and $\xi = 1$ reflects complete blockage of transmission.

**2.6.1 Scheme (I): Population-wide control.** Population-wide control measures include intervention measures for all individuals, such as wearing a facemask [74], routine sterilization [75], social distancing [76], 'work-from-home' policy [77], and mass vaccination programs.

Following [29], this control scheme (**I**) is expected to have the least efficacy in risk reduction and thus is treated as the baseline scenario.

In population-wide control measures, we consider that each individual reproduction number ($\lambda$) is reduced by a factor $\xi$ ($0 \le \xi < 1$) for fixed and variable components ($\lambda_F$ and $\lambda_V$), namely a relative reduction in the reproduction number. Then, on the population scale, the reproduction number ($R$) is also reduced by factor $\xi$, and thus the fixed and variable components become $(1 - \xi)R_F$ and $(1 - \xi)R_V$, respectively. The controlled reproduction is $(1 - \xi)R$. Thus, the PMF of offspring cases ($x$) generated by one seed case is the following Delaporte distribution, $f_D^{(1)}(x|\xi)$.

$$f_D^{(1)}(x|\xi) = \sum_{a=0}^{x} \left[ \frac{\Gamma(k+a)}{\Gamma(k)\Gamma(a+1)} \left( \frac{k}{(1-\xi)R_V + k} \right)^k \left( \frac{(1-\xi)R_V}{(1-\xi)R_V + k} \right)^a \cdot \frac{[(1-\xi)R_F]^{x-a} \cdot \exp(-(1-\xi)R_F)}{\Gamma(x-a+1)} \right]$$

The superscript '(1)' is merely for labeling purposes rather than powering.

For the final outbreak size ($c \ge 1$) generated by a single case under the control scheme (**I**), the PMF $\omega_D^{(1)}(c|\xi)$ can be derived as follows,

$$\omega_D^{(1)}(c|\xi) = \frac{1}{c} \sum_{a=0}^{c-1} \left[ \frac{\Gamma(kc+a)}{\Gamma(kc)\Gamma(a+1)} \left( \frac{k}{(1-\xi)R_V + k} \right)^{kc} \left( \frac{(1-\xi)R_V}{(1-\xi)R_V + k} \right)^a \cdot \frac{[(1-\xi)R_F c]^{c-a-1} \cdot \exp(-(1-\xi)R_F c)}{\Gamma(c-a)} \right]$$

which incorporated Eq (8) with $f_D^{(1)}(x|\xi)$.

**2.6.2 Scheme (II): High-risk-specific control.** High-risk-specific control measures target individuals with higher risk of superspreading potentials, e.g., individuals who frequently travel and contact others, and staff members sharing common facilities in the workplace. Thus, interventive measures such as city lockdowns and travel bans [78, 79], digital contact tracing at public places [80, 81], and gathering restrictions may interfere with the potential risks of spreading the disease by targeting high-risk individuals.

High-risk-specific control measures prioritize the variable component of the individual reproduction number ($\lambda_V$). Despite $\lambda_F$ being unchanged, the value of $\lambda_V$ is reduced so that individuals with higher risks of superspreading are less likely to achieve their potential for spreading diseases. To guarantee comparability with the population-wide control scheme, we maintain that controlled reproduction is $(1 - \xi)R$, and thus the value of $R_V$ reduces $\xi R$ units. Then, on the population scale, the reproduction number ($R$) is reduced by factor $\xi$. In the scenario that $\xi R > R_V$, equivalently $\xi > R_V / R = 1 - \rho$ or $\xi + \rho > 1$, the reduction will lead to $R_V = 0$, the remaining amount ($\xi R - R_V$) for the reduction is then passed to the fixed component $R_F$, and the Delaporte distribution reduces to the Poisson distribution with rate $R_F - (\xi R - R_V) = (1 - \xi)R$. Thus, the PMF of offspring cases ($x$) generated by one seed case is formulated as follows, $f_D^{(2)}(x|\xi)$.

$$f_D^{(2)}(x|\xi)$$
$$= \begin{cases} \sum_{a=0}^{x} \left[ \frac{\Gamma(k+a)}{\Gamma(k)\Gamma(a+1)} \left( \frac{k}{(R_V - \xi R) + k} \right)^k \left( \frac{(R_V - \xi R)}{(R_V - \xi R) + k} \right)^a \cdot \frac{R_F^{x-a} \cdot \exp(-R_F)}{\Gamma(x-a+1)} \right], & \text{for } \xi < 1 - \rho \\[2em] \frac{[(1-\xi)R]^x \cdot \exp(-(1-\xi)R)}{\Gamma(x+1)}, & \text{for } \xi \ge 1 - \rho \end{cases}$$

The superscript '(2)' is merely for labeling purposes instead of powering.

For the final outbreak size ($c \geq 1$) generated by a single case under the control scheme (**II**), the PMF $\omega_D^{(2)}(c|\xi)$ can be derived as follows,

$$\omega_D^{(2)}(c|\xi)$$
$$= \frac{1}{c} \begin{cases} \sum_{a=0}^{c-1} \left[ \frac{\Gamma(kc+a)}{\Gamma(kc)\Gamma(a+1)} \left( \frac{k}{(R_V - \xi R) + k} \right)^{kc} \left( \frac{(R_V - \xi R)}{(R_V - \xi R) + k} \right)^a \cdot \frac{(R_F c)^{c-a-1} \cdot \exp(-R_F c)}{\Gamma(c-a)} \right], & \text{for } \xi < 1 - \rho \\ \frac{(R_F c)^{c-1} \cdot \exp(-R_F c)}{\Gamma(c)}, & \text{for } \xi \geq 1 - \rho \end{cases}$$

which incorporated Eq (8) with $f_D^{(2)}(x|\xi)$.

In particular, when the Delaporte distribution is restricted to the NB distribution, the distributions $f_D^{(1)}(x|\xi)$ and $f_D^{(2)}(x|\xi)$ become equivalent. When $\xi = 0$, $f_D^{(1)}(x|\xi = 0) = f_D(x) = f_D^{(2)}(x|\xi = 0)$, and $\omega_D^{(1)}(c|\xi = 0) = \omega_D(c, 1) = \omega_D^{(2)}(c|\xi = 0)$.

**2.6.3 Risk outcome (I): Superspreading event.** The superspreading event is defined as the situation where an index case produces more secondary cases than the superspreading threshold ($y$). Following [29], when given $R$, the superspreading threshold $y$ is calculated as the 99th percentile of the Poisson distribution with rate $R$ [17]. Mathematically, $y$ satisfies $\mathbf{Pr}(X \leq y \mid X \sim \text{Poisson}(R)) = 0.99$. For example, with the reproduction number in the range from 1.5 to 3 for COVID-19 [41, 82–85], the superspreading threshold ($y$) ranges from 5 to 8 secondary cases.

Because $y$ can be determined for a given $R$, the risk of having a superspreading event is the probability that a seed case generates offspring cases equal to or greater than the superspreading threshold. When the control measures have no effect on reducing the reproduction number, i.e., $\xi = 0$, the risk of superspreading event $r_D$ is

$$r_D = 1 - \sum_{x=0}^{y-1} f_D(x)$$

Under control schemes (**I**) and (**II**), the risks of a superspreading event are as follows.

$$r_D^{(1)}(\xi) = 1 - \sum_{x=0}^{y-1} f_D^{(1)}(x|\xi), \text{ and } r_D^{(2)}(\xi) = 1 - \sum_{x=0}^{y-1} f_D^{(2)}(x|\xi),$$

respectively. Therefore, the control efficacies can be compared within or between control schemes given the same values of $R$ or $\xi$.

**2.6.4 Risk outcome (II): Large-scale outbreak.** A large-scale outbreak is defined as an outbreak with a final size ($c$) greater than 100, of which the threshold was adopted in [3, 29, 33]. Seeded by an index case, the final outbreak size $c$ ($\geq 1$) is modelled in Eq (8) and is translated into $h_D^{(1)}(c|\xi)$ and $h_D^{(2)}(c|\xi)$ under control schemes (**I**) and (**II**), respectively.

When $\xi = 0$, the risk of large-scale outbreak $r_D$ is

$$r_D = 1 - \sum_{c=1}^{100} \omega_D(c, 1)$$

Under control schemes (**I**) and (**II**), the risks of large-scale outbreak are

$$r_D^{(1)}(\xi) = 1 - \sum_{c=1}^{100} \omega_D^{(1)}(c|\xi), \text{ and } r_D^{(2)}(\xi) = 1 - \sum_{c=1}^{100} \omega_D^{(2)}(c|\xi),$$

respectively.

**2.6.5 Control efficacy.** To compare different control strategies, the relative reduction in risk or relative efficacy approach was adopted [35]. For overdispersed transmission, most infected individuals do not contribute to the expansion of the epidemic, the final size of the

outbreak could be drastically controlled by preventing relatively rare superspreading events [29]. Therefore, we measure the efficacy of control as the relative risk reduction (RRR) of having a superspreading event or leading to a large-scale outbreak in each seed case. As such, the following calculation applies to both risk outcomes (**I**) and (**II**).

Given $R$, the RRRs of control schemes (**I**) and (**II**) are

$$\text{RRR}^{(1)}(\xi) = 1 - \frac{r_{\text{D}}^{(1)}(\xi)}{r_{\text{D}}}, \text{ and } \text{RRR}^{(2)}(\xi) = 1 - \frac{r_{\text{D}}^{(2)}(\xi)}{r_{\text{D}}},$$

respectively. As such, both $\text{RRR}^{(1)}(\xi)$ and $\text{RRR}^{(2)}(\xi)$ should be interpreted as the control efficacy when there is a reduction in $R$ by factor $\xi$ against that there is no change in $R$.

For the comparison between two control schemes, the RRR of control scheme (**II**) against control scheme (**I**) is

$$\text{RRR}^{(2,1)}(\xi) = 1 - \frac{r_{\text{D}}^{(2)}(\xi)}{r_{\text{D}}^{(1)}(\xi)}$$

Specially, when $\rho = 0$, that is, under the NB framework, $\text{RRR}^{(1)}(\xi)$ and $\text{RRR}^{(2)}(\xi)$ are equal or $\text{RRR}^{(2,1)}(\xi) = 0$ for both risk outcomes (**I**) and (**II**).

We solved $\text{RRR}^{(2,1)}(\xi)$ as function of both $\rho$ and $\xi$ numerically for both outcomes with the dispersion $k$ fixed at 0.2 for COVID-19.

## 3 Results and discussion

By definition, the Delaporte distribution allows the decomposition of the individual reproduction number ($\lambda$) into two independent and additive components (i.e., $\lambda_{\text{F}}$ and $\lambda_{\text{V}}$). Although the offspring cases ($X_{\text{F}}$) generated from the $\lambda_{\text{F}}$ part are variable, the fixed component $\lambda_{\text{F}} = R_{\text{F}}$ is constant. In contrast, the variable component $\lambda_{\text{V}}$ is a Gamma-distributed variable that accounts for the differences between individual cases and shares the same definition and interpretation as in the NB distribution [29, 45]. As a generalization of the NB distribution, the Delaporte distribution appears different from the Poisson and NB distributions given the same mean $R$ and dispersion $k$ (see Fig 1), which is due to the effect of the additional parameter $\rho$. The term $\rho$ quantifies the fraction of the mean reproducibility that is fixed (or the same) across different cases. The classic NB model restricted the fixed (baseline) fraction $\lambda_{\text{F}}$ to be 0, indicating that there must be a proportion of individuals with (almost) 0 transmissibility, which appears unrealistic. Conversely, the Delaporte distribution allowed $\lambda_{\text{F}}$ to be a non-negative value, which is more flexible for complex situations. Theoretically, a lower value of either $\rho$ or $k$ indicates a higher scale of variability in individual infectiousness [29], that is, variance in the distribution of offspring. With other parameters fixed, a smaller $\rho$ leads to a larger (smaller) proportion of the most infectious primary cases ($P$) that produce the most (zero) secondary cases (Figs 2 and 3). The consistent negative relationship between $\rho$ and superspreading potential was demonstrated, and this relationship appears stronger as $k$ decreases. The most heterogeneous transmission occurs when both $k$ and $\rho$ are small, and the Delaporte distribution approaches the NB distribution. With the same $R$ and $k$, the Lorenz curve of the Delaporte distribution falls between those of the Poisson and NB distributions (Fig 4), where the position of the Delaporte distribution depends on $\rho$.

Fitting to several datasets of offspring (or secondary) cases, our estimates of NB parameters were consistent with previous studies (Table 1). When the $R_{\text{F}}$ estimate was greater than 0 for the Delaporte distribution, the dispersion $k$ estimate became greater than the $k$ estimate of the NB distribution. We found that the Delaporte distribution led to an improved or equivalent fitting performance compared to the NB distribution in terms of AIC values. The

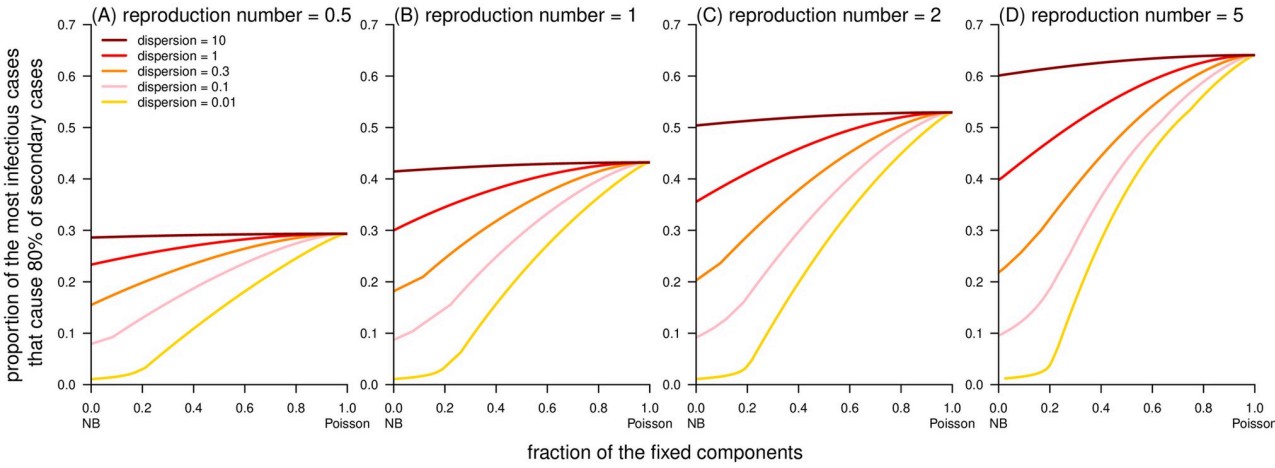

**Fig 2. Simulation results of the proportion ($P$) of the most infectious cases that cause ($Q =$) 80% of secondary cases as a function of fraction of fixed component ($\rho$) generated from Delaporte distributions.** The 'NB' in the horizontal axis label stands for negative binomial (distribution).

improvement in fitting performance was also reflected by the estimates of $R_F$, or equivalently $\rho$ (not shown as the main result). When the sample size is large, for example, datasets #1-#3, the Delaporte distribution has a higher goodness-of-fit in terms of likelihood values. The Delaporte distribution more accurately captures the observed offspring data than the NB distribution (Fig 5). In datasets #1-#3, the high-density regions of posterior distributions of $\rho$ were roughly skewed from 0.1 to 0.5. However, the improvement in explaining the real-world dataset becomes weak, or even not evident as sample size decreases, for example, datasets #4 and #5, where the NB distribution also yields satisfactory fitting performance. For datasets #2b and #2a, collected from 19 January to 19 April 2020 and from 20 April to 16 October 2020, respectively. It is worth noting that the estimated medians of $\rho$ increased from 0.21 to 0.56, while $k$ only had minor changes. With the same scales of $k$ and $R$, the increase in $\rho$ would lead to a decrease in the overdispersiveness of disease transmission, as well as a reduction in the risk of

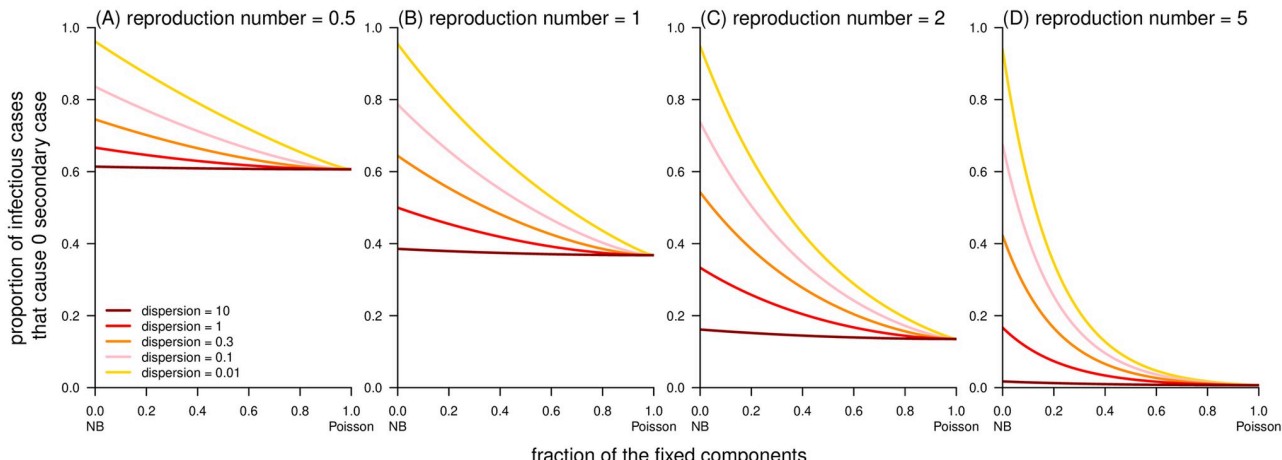

**Fig 3. Simulation results of the proportion of cases, i.e., $f_D(0)$, that cause 0 secondary case as a function of fraction of fixed component ($\rho$) generated from Delaporte distributions.** The 'NB' in the horizontal axis label stands for negative binomial (distribution).

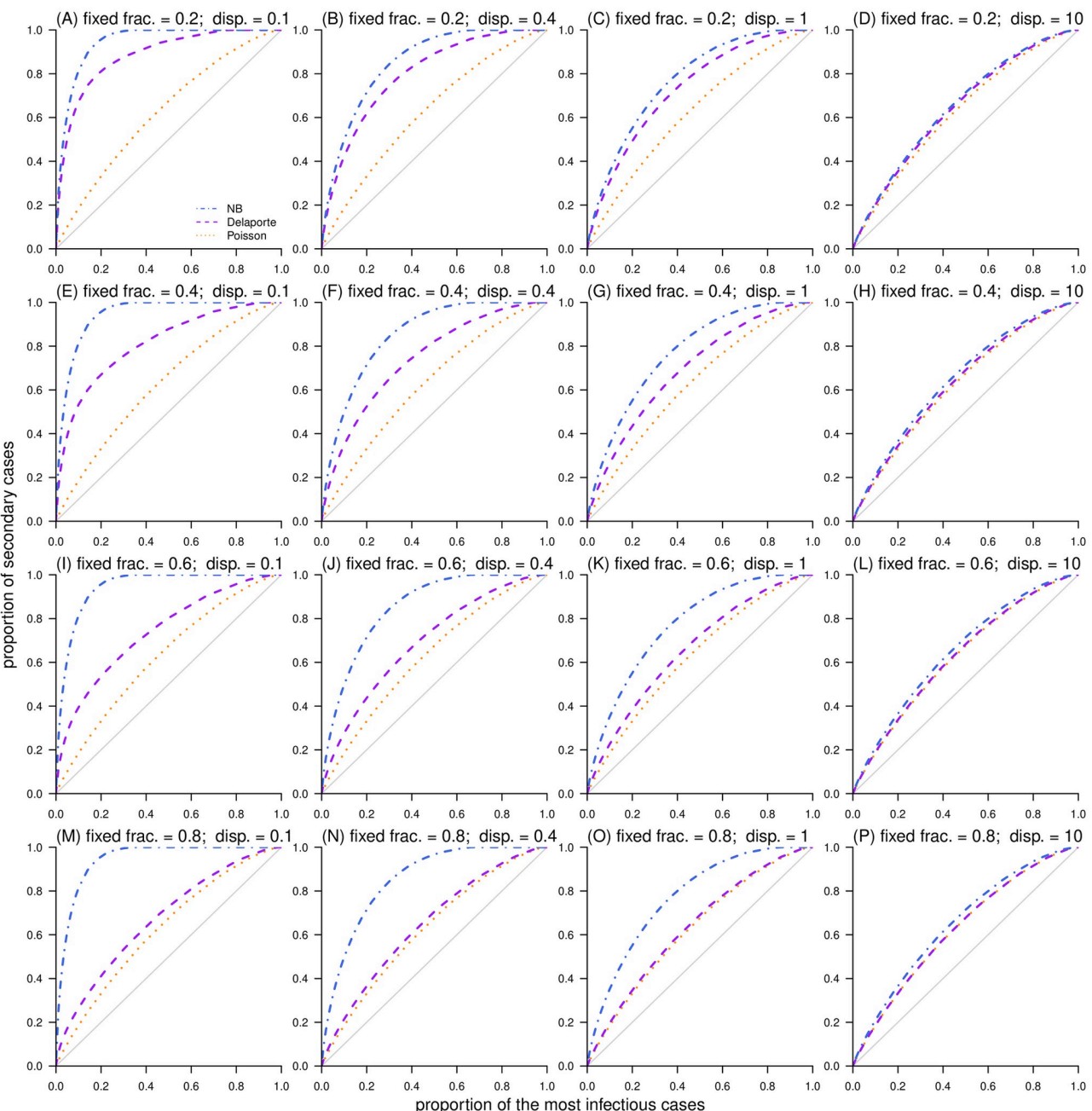

**Fig 4. Simulation results of the expected proportion of secondary cases ($Q$) due to the proportion of the most infectious cases ($P$), i.e., Lorenz curve, generated from Poisson (in orange), negative binomial (in blue), and Delaporte (in purple) distributions.** In each panel, the diagonal line shows the scenario of perfect homogeneity (i.e., uniform distribution). In each panel label, 'fixed frac.' is the fraction of fixed component ($\rho$), and 'disp.' is the dispersion parameter ($k$).

superspreading. This finding was consistent with the conclusion in [33], which also discussed the impact of various local nonpharmaceutical interventions on the transmission characteristics of COVID-19 in South Korea.

**Table 1. The summary of parameter estimates of offspring distribution in the existing literature and this study.** The '−2·log(L)' denotes twice of the negative log-likelihood. The highlighted estimates are considered as main results for Delaporte distribution (in red) and negative binomial (NB) distribution (in blue).

| disease | dataset | | model distribution | reproduction number | components of Poisson rate | | | fitting performance | | reference of estimation |
|---|---|---|---|---|---|---|---|---|---|---|
| | label | source | | | fixed | variable | | −2·log (L) | AIC | |
| | | | | | | mean | dispersion | | | |
| COVID-19 | (#1) | Xu et al. [12] (n = 2214) | Poisson | equals the fixed component | 0.69 (0.65, 0.72) | none | | 5137.46 | 5139.46 | this study |
| | | | negative binomial | equals the mean of variable component | none | 0.69 (0.62, 0.77) | 0.70 (0.59, 0.98) | not reported | | He et al. [30] |
| | | | | | | 0.69 (0.64, 0.74) | 0.74 (0.62, 0.89) | 4658.85 | 4662.85 | this study |
| | | | Delaporte | 0.69 (0.64, 0.74) | 0.26 (0.16, 0.33) | 0.43 (0.35, 0.54) | 0.24 (0.15, 0.40) | 4635.37 | 4641.37 | this study |
| | (#2a) | Lim et al. [33] (n = 1401) | Poisson | equals the fixed component | 0.68 (0.64, 0.72) | none | | 3486.90 | 3488.90 | this study |
| | | | negative binomial | equals the mean of variable component | none | not reported | 0.85 (0.70, 1.05) | not reported | | Lim et al. [33] |
| | | | | | | 0.68 (0.62, 0.74) | 0.85 (0.70, 1.06) | 3175.24 | 3179.24 | this study |
| | | | Delaporte | 0.68 (0.62, 0.75) | 0.38 (0.30, 0.45) | 0.30 (0.22, 0.40) | 0.11 (0.06, 0.20) | 3134.44 | 3140.44 | this study |
| | (#2b) | Lim et al. [33] (n = 344) | Poisson | equals the fixed component | 0.81 (0.71, 0.90) | none | | 1234.96 | 1236.96 | this study |
| | | | negative binomial | equals the mean of variable component | none | not reported | 0.23 (0.15, 0.28) | not reported | | Lim et al. [33] |
| | | | | | | 0.81 (0.64, 1.06) | 0.23 (0.17, 0.30) | 764.62 | 768.62 | this study |
| | | | Delaporte | 0.81 (0.61, 1.18) | 0.17 (0.03, 0.27) | 0.65 (0.43, 1.00) | 0.09 (0.05, 0.19) | 751.66 | 757.66 | this study |
| | (#3) | Adam et al. [17] (n = 290) | Poisson | equals the fixed component | 0.58 (0.50, 0.68) | none | | 699.85 | 701.85 | this study |
| | | | | | 0.58 (0.50, 0.69) | | | 699.85 | 701.85 | Adam et al. [17] |
| | | | negative binomial | equals the mean of variable component | none | 0.58 (0.45, 0.72) | 0.43 (0.29, 0.67) | 589.93 | 593.93 | |
| | | | | | | 0.58 (0.45, 0.73) | 0.43 (0.29, 0.63) | 589.92 | 593.92 | this study |
| | | | Delaporte | 0.59 (0.46, 0.78) | 0.17 (0.04, 0.30) | 0.42 (0.25, 0.63) | 0.16 (0.06, 0.40) | 585.80 | 591.80 | this study |
| | (#4) | Zhang et al. [19] (n = 47) | Poisson | equals the fixed component | 0.71 (0.49, 1.01) | none | | 126.42 | 128.42 | this study |
| | | | negative binomial | equals the mean of variable component | none | 0.67 (0.54, 0.84) | 0.25 (0.13, 0.88) | not reported | | Zhang et al. [19] |
| | | | | | | 0.71 (0.39, 1.77) | 0.28 (0.10, 0.80) | 95.35 | 99.35 | this study |
| | | | Delaporte | 0.72 (0.36, 1.70) | 0.00 (0.00, 0.08) | 0.72 (0.34, 1.69) | 0.23 (0.10, 0.54) | 95.21 | 101.21 | this study |
| SARS | (#5) | Shen et al. [5] (n = 34) | Poisson | equals the fixed component | 1.76 (1.37, 2.24) | none | | 274.88 | 276.88 | this study |
| | | | negative binomial | equals the mean of variable component | none | 1.88 (0.41, 3.32) | 0.12 (0.08, 0.42) | not reported | | Lloyd-Smith et al. [29] |
| | | | | | | 1.96 (0.67, 4.37) | 0.10 (0.02, 0.19) | 78.78 | 82.78 | this study |
| | | | Delaporte | 2.07 (0.52, 3.23) | 0.06 (0.00, 0.31) | 2.00 (0.51, 3.01) | 0.05 (0.01, 0.17) | 78.18 | 84.19 | this study |

*Note*: All parameter estimates were summarized in the 'median (95% credible interval)' of posterior distribution format.

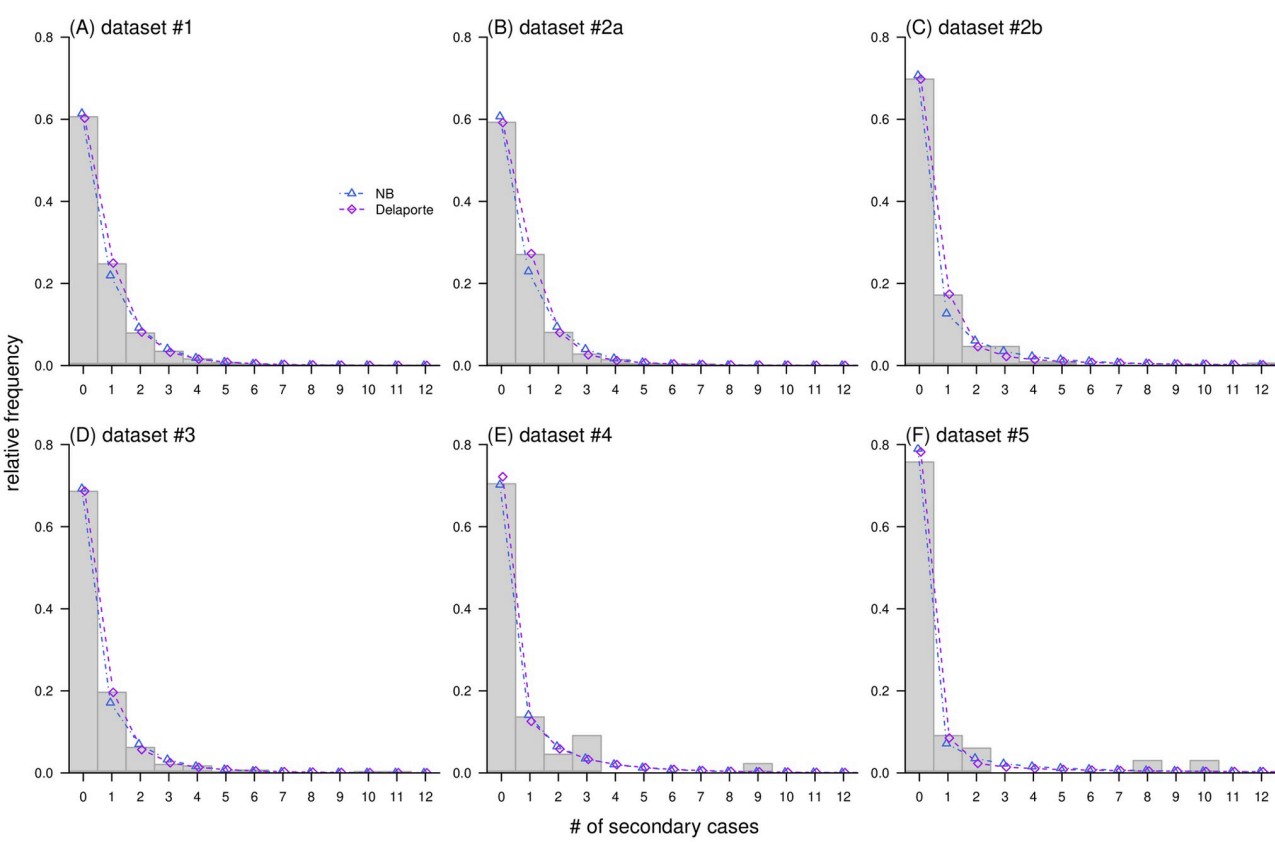

**Fig 5. Fitting results of offspring distributions using the medians of posterior distributions for model parameters.** In each panel, probability mass functions (PMF) of negative binomial (NB, in blue), and Delaporte (in purple) distributions are shown in dots and lines, and the observations of number of secondary cases per infector (in grey) are in histogram. *Note*: The PMFs of NB and Delaporte distributions were shifted horizontally in each panel with slight jitters at −0.05 and +0.05, respectively to aid visualization and comparison.

The likelihood ratio (LR) test has been proposed for model selection between the NB and the Delaporte distributions [11, 66], and yields satisfactory testing performance. We found an increasing statistical power of the LR test for identifying the improvement of Delaporte distribution as the sample size increased. The simulation results of the testing power show consistent trends as observed in datasets #1-#5 (Fig 6A). To secure a power larger than 0.80, surveillance may require a sample size above 400, see Fig 6B. Although the type I error rate appears slightly high around 0.03 when sample size ranges from 100 to 300 (Fig 6D–6E), while the type I error rate is generally conservative for a wide range of sample sizes from 30 to 3000 (Fig 6F). Similar non-monotone trends of the type I error rate have also been previously reported for other testing purposes [40]. The testing performance of increasing power and conservative type I error suggest that the LR test is informative in capture the true characteristics of over-dispersed offspring distribution with a low chance of false alarms.

In practical analysis, one may also be interested in obtaining estimators for $R$ and $k$ given the parameter estimates of the Delaporte distribution. Because the closest theoretical formula may be complex to derive, a convenient approximation using moments of the Delaporte distribution could be considered. To distinguish the dispersion parameters, we denote $k_{NB}$ and $k_D$ for the NB and Delaporte distributions, respectively. For a given Delaporte distribution, the first moment (i.e., mean) is $R_F + R_V$, and the second central moment (i.e., variance) is

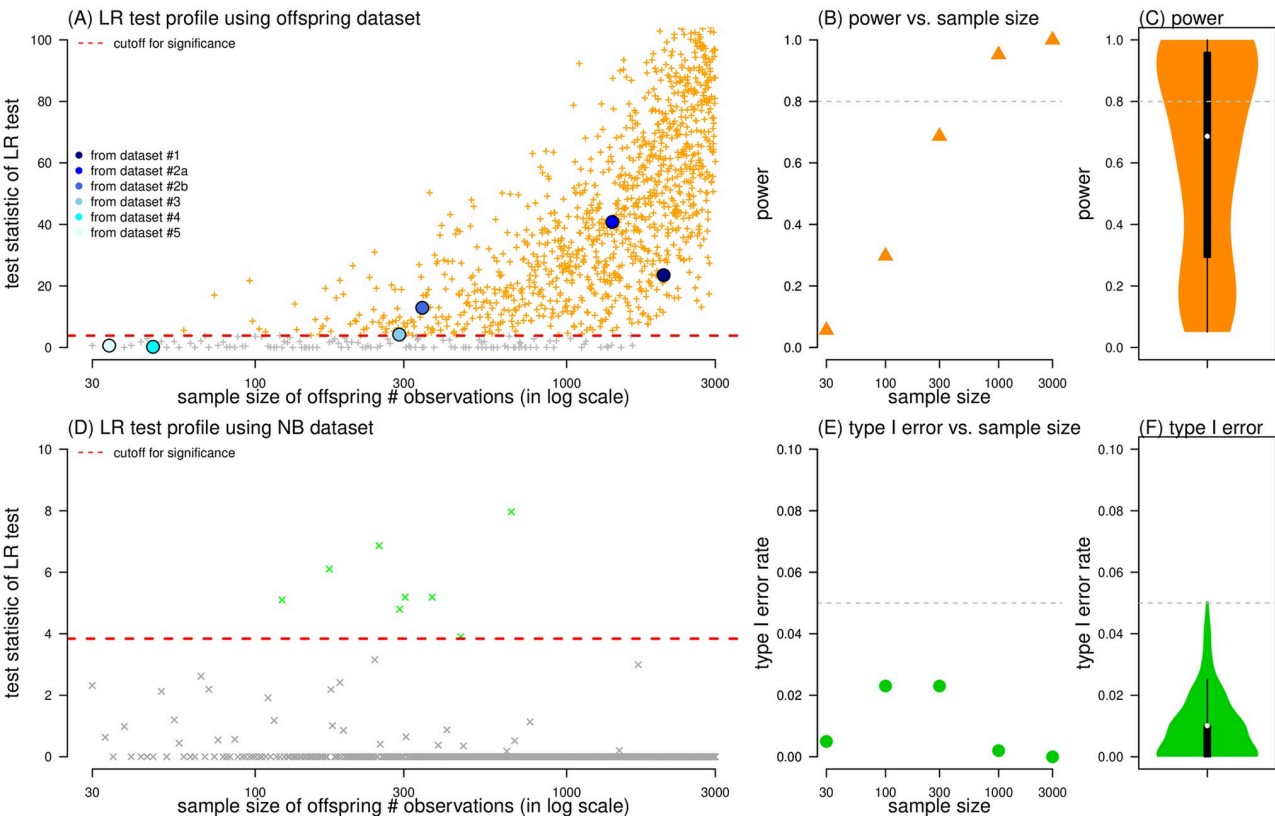

**Fig 6. The power and type I error rate of the likelihood ratio (LR) test for Delaporte distribution against negative binomial (NB) distribution.** Panels (A) and (D) show the test statistics (dots) from LR test, and the critical threshold (red horizontal dashed line) for *p*-value < 0.05. In panel (A), the '+' dots are 10000 pseudo datasets generated by random sampling with replacement from the real-world datasets, and the circle dots represent datasets #1-#5. Panels (B) and (E) summarized the power and type I error rate of LR test for Delaporte distribution against NB distribution as a function of sample size. Panels (C) and (F) summarized the power and type I error rate of LR test with sample size reciprocal-distributed from 30 to 3000. In panel (D), the '×' dots are generated by 10000 datasets generated by Monte Carlo sampling from NB distributions. In panels (B) and (C), the horizontal dashed line is the threshold of power at 0.80. In panels (E) and (F), the horizontal dashed line is the threshold of type I error rate at 0.05.

$R_\mathrm{F} + R_\mathrm{V}\left(1 + \frac{R_\mathrm{V}}{k_\mathrm{D}}\right)$. Thus, if let the NB distribution have the same value of mean and variance, for the approximated NB distribution, we have $\widehat{R} = R_\mathrm{F} + R_\mathrm{V}$, and

$\widehat{k_\mathrm{NB}} = \left(\frac{\widehat{R}}{R_\mathrm{V}}\right)^2 \cdot k_\mathrm{D} = \left(\frac{R_\mathrm{F}+R_\mathrm{V}}{R_\mathrm{V}}\right)^2 \cdot k_\mathrm{D} = \frac{k_\mathrm{D}}{(1-\rho)^2}$. Although this approximation can be directly calculated rapidly, by using the estimates of the example offspring datasets, we note that $\widehat{k_\mathrm{NB}}$ here appears slightly lower than the posterior estimates of $k_\mathrm{NB}$ in Table 1.

The real-world datasets adopted in this study were offspring cases per seed case observations, but more generally, the Delaporte distribution can be extended to describing one-generation cluster or final outbreak size observations. For the one-generation cluster size $j$ distribution, we derived that $h_\mathrm{D}(j|i)$ also follows a Delaporte distribution with parameters not only determined on the original parameter set of $f_\mathrm{D}(X)$ but also by the number of seed cases $i$. Specifically, $f_\mathrm{D}(X)$ can be translated into $h_\mathrm{D}(j|i)$ by multiplying parameters $\rho$ and $R$ by $i$, see Eq (6). A previous study determined that one-generation cluster size follows a NB distribution $h_\mathrm{NB}(j|i)$ under the NB-distributed offspring assumption [40], which is similar to our extension of this finding to the situation of the Delaporte distribution. To assess the impact of $\rho$ on

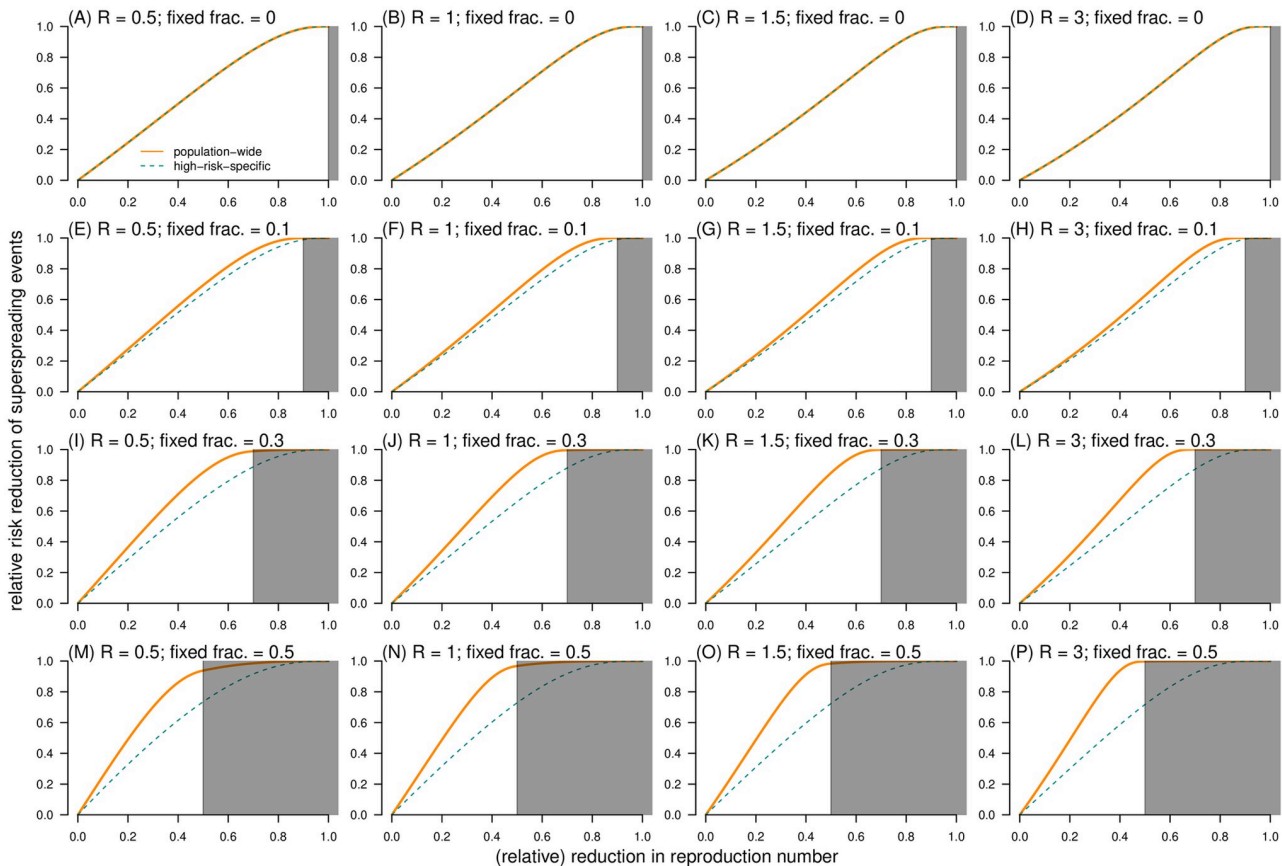

**Fig 7. The relative risk reduction (RRR) of outcome (I): Having superspreading event as a function of the relative reduction in reproduction number ($\xi$).** The RRR of control scheme (**I**) RRR$^{(1)}$($\xi$) is dashed cyan curve, and the RRR of control scheme (**II**) RRR$^{(2)}$($\xi$) is bold orange curve. In each panel, the dispersion parameter $k$ is fixed at 0.2, and the shading region indicates the situation that $\xi \geq 1 - \rho$. In each panel label, '$R$' is the reproduction number, and 'fixed frac.' is the fraction of fixed component ($\rho$).

disease outbreaks, the final outbreak size $c$ distribution can be used to evaluate pandemic potentials seeded by $i$ source (or imported) cases [2, 38, 73, 86]. Thus, $\omega_D(c, i)$was derived in Eq (7), and appeared to be an extension of the NB version $\omega_{NB}(c, 1)$ in [3, 33], see the special case of Eq (8).

To illustrate the translation from the final outbreak size probability in Eq (7) to the likelihood-based estimation, we adopted the final outbreak size observations of the Middle East respiratory syndrome coronavirus (MERS-CoV infection in the Middle East region, which was reported in [87]. The dataset has a sample size of 55 outbreaks, including a total of 104 laboratory confirmed MERS cases, and all final outbreaks were seeded by single cases, as also summarized and studied in [3]. Hence, Eq (8) was used to construct the likelihood function for the Delaporte distribution. We estimated $R_F$ at 0.17 (95%CrI: 0.03, 0.45), $R_V$ at 0.32 (95% CrI: 0.01, 1.53), and $k$ at 0.04 (95%CrI: 0.00, 0.19) with an AIC of 114.60. We also repeated the estimation using the NB distribution, which leads to $R$ at 0.47 (95%CrI: 0.30, 0.78) and $k$ at 0.27 (95%CrI: 0.10, 0.98) with an AIC of 115.68. For the previous estimates using NB in [3], it was estimated that $R$ was 0.47 (95%CrI: 0.29, 0.80) and $k$ was 0.26 (95%CrI: 0.09, 1.24), which was in line with our estimates. The $k$ estimate appears lower in the Delaporte distribution, and the $\rho$ estimate at 0.33 (95%CrI: 0.05, 0.98) was greater than 0, thus the fixed part of $R$ was evident, which was also indicated by the difference in the AIC values.

Aside from the impact of $k$ in determining the probability of risk outcome (**I**): in super-spreading events, as described in [3, 29], the parameter $\rho$ also has an similar impact, and further influences the efficacy of different control strategies. With the same among ($\xi$) of reduction in $R$, the control efficacies (RRR) of both population-wide and high-risk-specific control schemes increased with $\xi$ (Fig 7). To compare the two control schemes, we found that the control scheme (**II**) has a higher control efficacy than scheme (**I**) in terms of the RRR of superspreading event, i.e., $RRR^{(2,1)}(\xi)$. Effective control efforts may allow us to anticipate highly infectious source cases or the contexts in which a seed case may likely expose many susceptible individuals in advance. Then, the scale of the variable component of the reproduction number was reduced efficiently under the control scheme (**II**), such that a substantial proportion of superspreaders can be controlled. With $\xi < 1 - \rho$, the general (or linear) tendency of $RRR^{(2,1)}(\xi)$ increased rapidly as $\xi$ or $\rho$ increased (Fig 8). The largest value of $RRR^{(2,1)}(\xi)$ can be reached when $\xi$ is close (but not necessarily approaching) to $1 - \rho$. When $\rho = 0$, we illustrated that $RRR^{(1)}(\xi) = RRR^{(2)}(\xi)$ (Fig 7A–7D), which indicated that $RRR^{(2,1)}(\xi) = 0$. In other words, with the effects of $\rho$ ($> 0$), the outperformance of high-risk-specific control scheme may become evident in terms of achieving $RRR^{(2,1)} > 0$ for some values of $\xi$ (Fig 8).

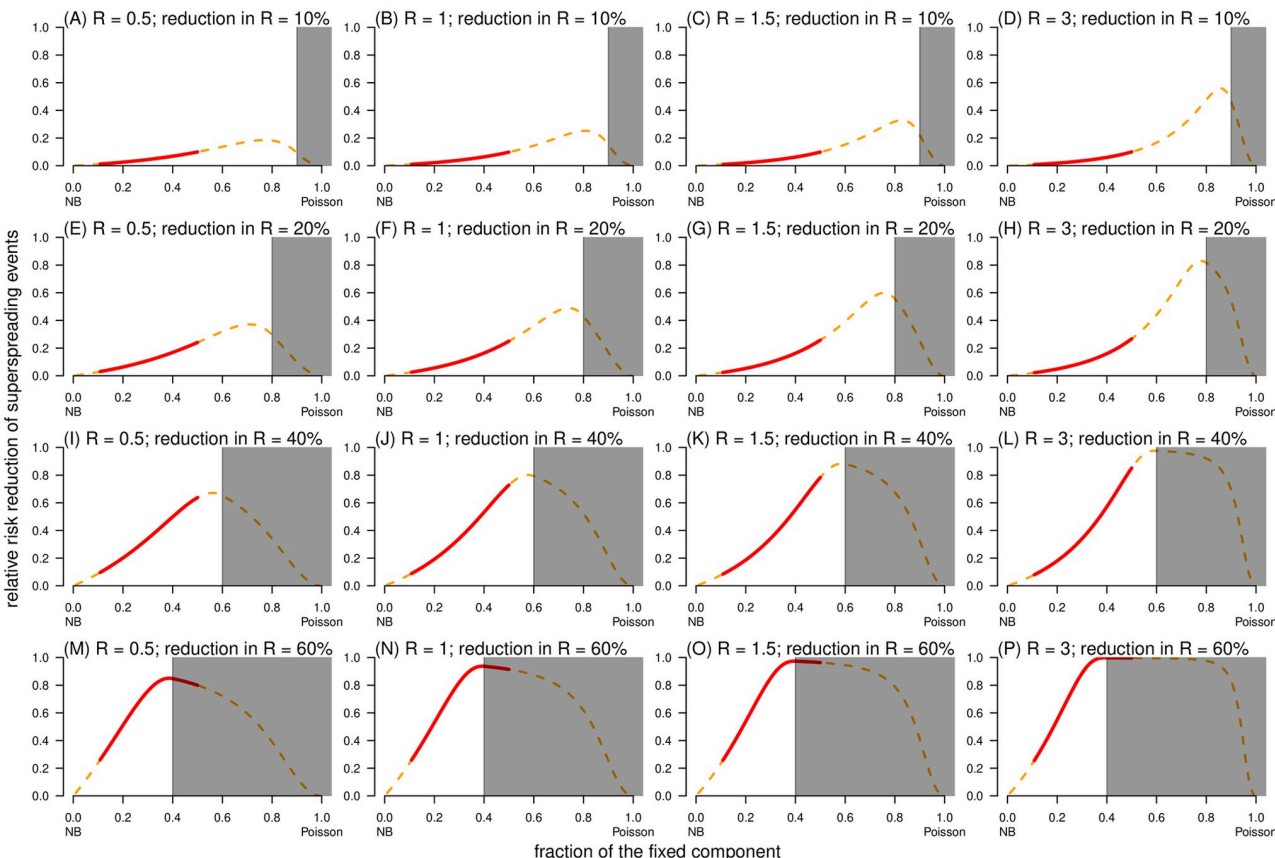

**Fig 8. The relative risk reduction, $RRR^{(2,1)}(\xi)$, of outcome (I): Having superspreading event under control scheme (II) against scheme (I) as a function of the fraction of fixed component ($\rho$).** In each panel, the dispersion parameter $k$ is fixed at 0.2, the shading region indicates the situation that $\xi \geq 1 - \rho$, and the bold red segment highlights the range of $\rho$ from 0.1 to 0.5, which characterizes the feature of COVID-19. In each panel label, '$R$' is the reproduction number, and 'reduction in $R$' is the relative reduction in reproduction number ($\xi$). The 'NB' in the horizontal axis label stand = s for negative binomial (distribution).

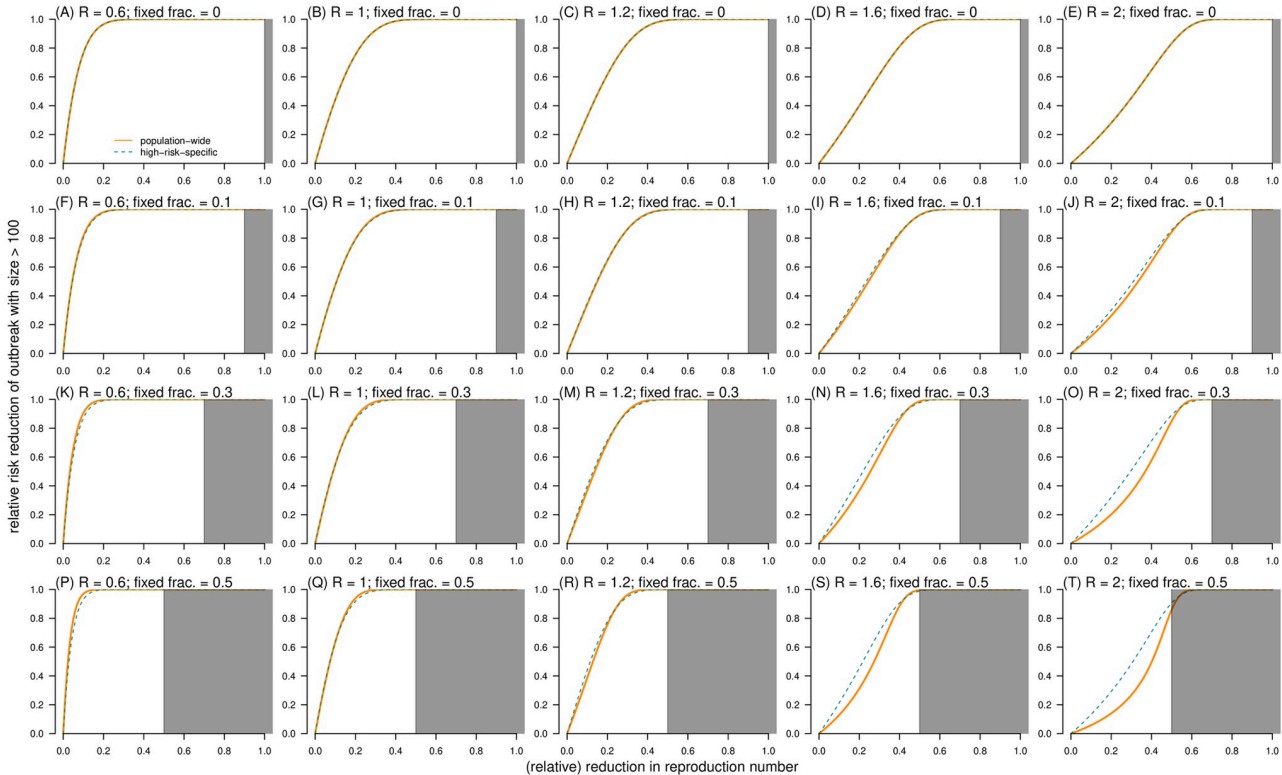

**Fig 9. The relative risk reduction (RRR) of outcome (II): Having outbreak with final size $c > 100$ as a function of the relative reduction in reproduction number ($\xi$).** The RRR of control scheme (**I**) RRR$^{(1)}(\xi)$ is dashed cyan curve, and the RRR of control scheme (**II**) RRR$^{(2)}(\xi)$ is bold orange curve. In each panel, the dispersion parameter $k$ is fixed at 0.2, and the shading region indicates the situation that $\xi \geq 1 - \rho$. In each panel label, '$R$' is the reproduction number, and 'fixed frac.' is the fraction of fixed component ($\rho$).

For effective control strategies aiming to reduce the risk of outcome (**II**): large-scale outbreak, the RRR was determined by $\xi$, $\rho$, and $R$. Consistent with the trends of risk outcome (**I**) in Fig 7, a large-scale outbreak was less likely to occur as $\xi$ increased despite control schemes (Fig 9). When $\rho = 0$, we illustrated that RRR$^{(1)}(\xi) = $ RRR$^{(2)}(\xi)$, see Fig 9A–9E, which indicated RRR$^{(2,1)}(\xi) = 0$. Unlike SSE, the population-wide control scheme outperformed the high-risk-specific control scheme with RRR$^{(2,1)} < 0$ when $R$ was large and $\xi$ was small, but the direction (or sign) may change to RRR$^{(2,1)} > 0$ for small $R$ or large $\xi$ (Fig 10). On one hand, the high-risk-specific control scheme was more effective in reducing the outbreak risks under subcritical transmission. In self-limited (or stuttering) outbreak, although SSEs rarely occur, they have a significant contribution to the expansion of transmission [57]; thus, the risk of outbreak can be drastically reduced by targeting high-risk individuals [36]. On the other hand, this implied that when the epidemic curve is growing in terms of reproduction numbers larger than 1, a substantial proportion of transmission is due to the fixed part ($\lambda_F = R_F$) of individual infectiousness, that is, subspreading events [88]. Despite the variable part $R_V$, a large $R_F$ results in stable reproducibility of infections, and RRR$^{(2,1)} < 0$ with a moderate scale of $\rho$ (from 0.1 to 0.5 for COVID-19) (Fig 10T). Therefore, population-wide interventions may successfully control disease transmission on a general scale before the implementation of high-risk-specific control strategies subsequently.

Conversely, under extremely intensive control measures in terms of $\xi \to 1$, the chance of large-scale outbreak diminishes despite different control schemes. For example, mainland

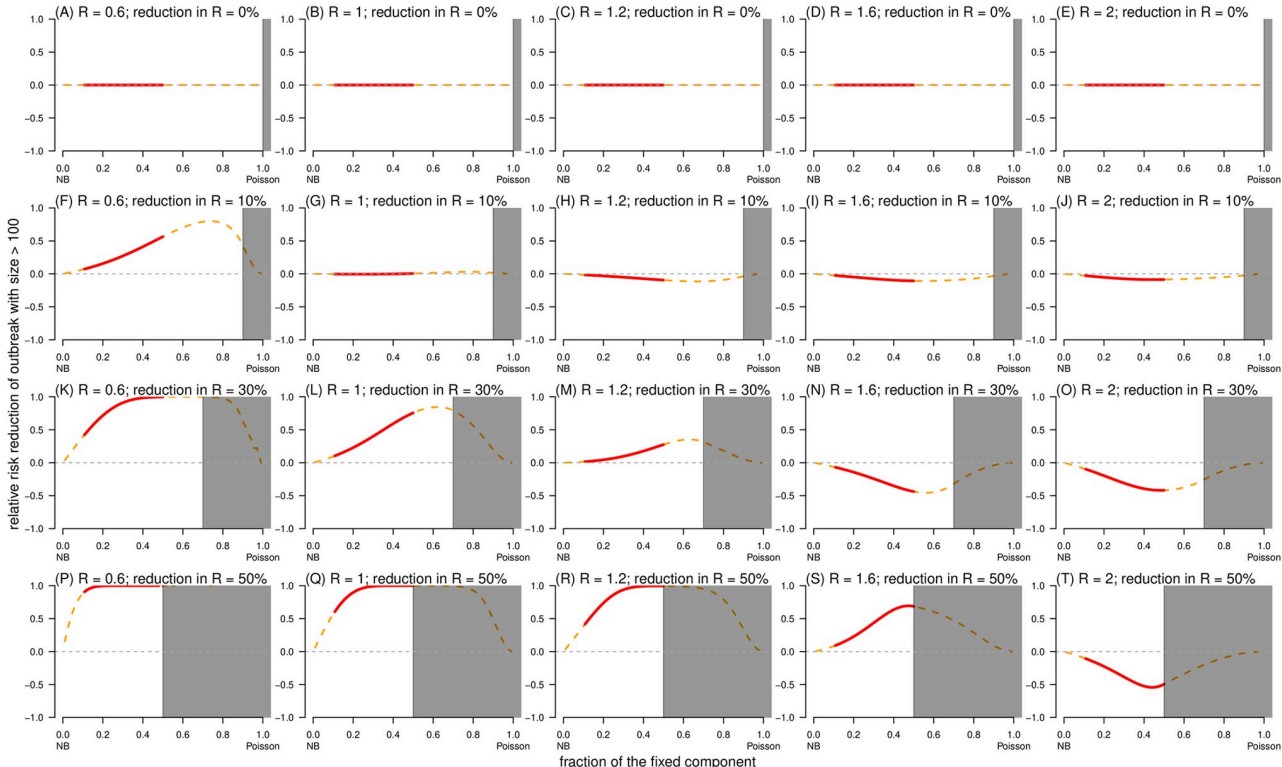

**Fig 10. The relative risk reduction, RRR$^{(2,1)}$($\xi$), of outcome (II): Outbreak with final size $c > 100$ under control scheme (II) against scheme (I) as a function of the fraction of fixed component ($\rho$).** In each panel, the dispersion parameter $k$ is fixed at 0.2, the shading region indicates the situation that $\xi \geq 1 - \rho$, and the bold red segment highlights the range of $\rho$ from 0.1 to 0.5, which characterizes the feature of COVID-19. In each panel label, '$R$' is the reproduction number, and 'reduction in $R$' is the relative reduction in reproduction number ($\xi$). The horizontal dashed grey line marked the level of RRR = 0. The 'NB' in the horizontal axis label stand = s for negative binomial (distribution).

China has achieved satisfactory COVID-19 control outcomes [89]. Although Chinese authorities relaxed population-wide policies in recent months, high-risk-specific control measures secured intensive and compulsory digital contact tracing efforts to monitor the risks of infection at the level of an individual's daily routine [90, 91]. In our theoretical framework, this indicates a high value of $\xi$ for control scheme (**II**), which leads to a remarkably low risk of outbreaks (Fig 9).

This study has limitations. First, although the Delaporte distribution is a theoretical generalization of the NB distribution, our data analysis focused on determining whether there is statistical evidence supporting the improvement in fitting performance without investigating the mechanistic side of the decomposition of the reproduction number. For example, population-level factors such as contact size and frequency (e.g., household size) [25], and heterogeneity of population density, or individual-level factors such as biological determinants (e.g., evolutionary adaptation and in-host viral kinetics) [92, 93], behavioral or social factors [32], and lifestyle habits might contribute to establishing superspreading potentials [29, 40]. Second, with regard to the parameter estimation part, we assumed that all offspring observations were accurately reported without selection bias, which might not always be acceptable [85, 94–97]. In cases of considerable reporting or selection bias, adjustments on statistical inference can resolve such issues to some extent by modifying the likelihood framework, for example, by truncation and compounding [11, 46, 57]. Lastly, for the evaluation of control effects, although the final

outbreak size ($c$) distribution was formulated under two schemes, we failed to find an analytical form for the condition with respect to $R$ and $\xi$, such that $RRR^{(2,1)} > 0$ or otherwise. Instead, we performed numerical simulations to check the sign of $RRR^{(2,1)}$ (shown visually in Fig 10), regarding the most feasible parameter ranges of COVID-19. Hence, the Delaporte distribution needs to be considered as a tool to monitor the three parameters to understand the transmission characteristics of infectious diseases and to provide information for strategic decision-making processes involving control measures.

In summary, as a generalization of the classic NB distribution, the Delaporte distribution can be adopted to decompose the reproduction number from the individual level to the population level and to characterize the transmission of infectious disease. The Delaporte distribution demonstrates statistical improvement in fitting the distributions of the real-world offspring cases' distributions against the NB distribution, and it presents increasing power and conservative type I error rates in detecting such an improvement in the goodness-of-fit with the LR test. Numerical simulation illustrated that the three parameters of the Delaporte distribution are important in understanding disease transmission characteristics and for advising of appropriate control strategies and providing new insights distinct from the NB model.

## Declarations

**Ethics approval and consent to participate.** The COVID-19 contact tracing data were obtained from literature, which were originally collected via the public domains, and thus neither ethical approval nor individual consent was applicable.

## Author Contributions

**Conceptualization:** Shi Zhao.

**Data curation:** Shi Zhao.

**Formal analysis:** Shi Zhao.

**Funding acquisition:** Daihai He, Maggie H. Wang.

**Investigation:** Shi Zhao.

**Methodology:** Shi Zhao, Marc K. C. Chong, Mu He, Daihai He.

**Project administration:** Shi Zhao.

**Resources:** Shi Zhao, Sukhyun Ryu.

**Software:** Shi Zhao.

**Supervision:** Maggie H. Wang.

**Validation:** Shi Zhao, Boqiang Chen.

**Visualization:** Shi Zhao.

**Writing – original draft:** Shi Zhao.

**Writing – review & editing:** Marc K. C. Chong, Sukhyun Ryu, Zihao Guo, Mu He, Boqiang Chen, Salihu S. Musa, Jingxuan Wang, Yushan Wu, Daihai He, Maggie H. Wang.

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
