## [Decision Letter · Decision Letter 0]

28 Feb 2022

Dear Mr. Zhao,

Thank you very much for submitting your manuscript "Characterizing superspreading potential of infectious disease: Decomposition of individual transmissibility" for consideration at PLOS Computational Biology.

As with all papers reviewed by the journal, your manuscript was reviewed by members of the editorial board and by several independent reviewers. In light of the reviews (below this email), we would like to invite the resubmission of a significantly-revised version that takes into account the reviewers' comments.

Both Reviewers have raised excellent points that should be helpful to the authors to strengthen their work.

We cannot make any decision about publication until we have seen the revised manuscript and your response to the reviewers' comments. Your revised manuscript is also likely to be sent to reviewers for further evaluation.

Sincerely,

Gerardo Chowell, PhD

Guest Editor

PLOS Computational Biology

Nina Fefferman

Deputy Editor

PLOS Computational Biology

Both Reviewers have raised excellent points that should be helpful to the authors to strengthen their work.

Reviewer's Responses to Questions

**Comments to the Authors:**

Reviewer #1: Please see the attachment.

Reviewer #2: This paper provides an interesting viewpoint on the potential use of Delaporte distribution for the offspring distribution as an extended alternative to the negative-binomial distribution. The explanation of the methods is clear and well supported. Inclusion of not only offspring distribution data but also single-generation and final cluster size data is very useful. Moreover, some of the study findings, including that the temporal change in South Korean data is explained by the change in the R_F/R_V proportion (rho), not in k, are of epidemiological interest. This may bring a new insight into how we may understand the offspring distribution from outbreaks under interventions.

Overall I feel the manuscript is of good quality and do not have much major criticism. Meanwhile, I would like to list two suggestions which I believe could further improve the epidemiological relevance of this paper; if the authors believe these do not make enough sense or involve overwhelming amount of additional effort (which is not what I wish to impose), I would be happy to be convinced.

- While some may adopt the Delaporte distribution for characterizing the offspring distribution, I expect that the majority of studies might continue to use the negative-binomial distribution (because it’s more mathematically convenient). In that case, it might be important to ensure that the interpretation of the estimates from these distributions are reasonably linked. Specifically, a study may fit a negative binomial distribution to a dataset (let’s call it dataset A) which is actually better represented by a Delaporte distribution, and report estimates for negative-binomial R and k. If one wishes to analyse another dataset (dataset B) using a Delaporte distribution and compare the results with these estimates, it would be convenient to have a measure that quantifies the similarity between a negative-binomial and a Delaporte distribution. For example, the estimated distributions in Figure 5 are more or less very similar between the NB and Delaporte, however their estimates are very different (because they represent different quantities). If the Delaporte distribution is close enough to a negative binomial distribution, NB might be a practical choice to ensure comparability and/or mathematical convenience (even if AIC suggests Delaporte is preferred). It would be therefore nice to have the following statistics given the Delaporte parameters R_F, R_V, and k : (1) estimators for R and k of a negative distribution that is the most similar to the given Delaporte distribution; (2) a measure of the similarity between the Delarporte distribution and the negative distribution given by (1), or conditions for the NB approximation to work well. However, these may not be easily obtained from the Delaporte estimates (e.g., require nonlinear optimization), in which case I would not request it as strongly.

- If there is an appropriate dataset available, I would wish to see an application example of the Delaporte distribution to the final outbreak size c, in addition to the currently included offspring distribution data (e.g., what correspond to Fig 5 and Table 1).

Minor comments

- L122 addictive -> additive?

- L124 I believe the individual reproduction number here is rather a non-standard usage (e.g. see https:doi.org/10.1371/journal.pone.0000758) and thus I feel this needs to be redefined/explained.

- L163–4 can you add a bit more context such that it’s clear in what way the interpretation of k is consistent with NB?

- L176 I believe ⌊∙⌋ is the floor function, not rounding (and the floor function should be used here).

- Section 2.4: In this setting I believe AIC and LR test are equivalent (difference of 2 in AIC corresponds to p = 0.05 in LR test) and including both seems redundant (and AIC is more widely used because it’s a more general framework). Could the same conclusion be obtained if the AIC-based model selection was used in place of LR for the discussion of type I/II errors?

- L315 It -> It is?

- L354–357 This is an important point. Can you briefly mention where the assumption of R<1 was introduced (L334) that this assumption would be relaxed in the end? Otherwise the reader may misinterpret that the final cluster size likelihood can only be used for R<1.

- Figures: there are an overwhelming number of figures (with even many panels). Some of them may better fit in the supplementary file.

- Throughout, I suggest the authors always distinguish between one-generation cluster size and final outbreak size and try to avoid just saying “cluster size” or “outbreak size” for best clarity.

- I found some typos/grammatical errors in the manuscript (which I named above) and I suggest the authors proofread the manuscript once again.

**Have the authors made all data and (if applicable) computational code underlying the findings in their manuscript fully available?**

Reviewer #1: None

Reviewer #2: Yes

PLOS authors have the option to publish the peer review history of their article (what does this mean?). If published, this will include your full peer review and any attached files.

Reviewer #1: No

Reviewer #2: No
---

## [Decision Letter · Decision Letter 1]

6 Jun 2022

Dear Mr. Zhao,

We are pleased to inform you that your manuscript 'Characterizing superspreading potential of infectious disease: Decomposition of individual transmissibility' has been provisionally accepted for publication in PLOS Computational Biology.

Best regards,

Gerardo Chowell, PhD

Guest Editor

PLOS Computational Biology

Nina Fefferman

Deputy Editor

PLOS Computational Biology

Jason A. Papin

Editor-in-Chief

PLOS Computational Biology

Feilim Mac Gabhann

Editor-in-Chief

PLOS Computational Biology

Reviewer's Responses to Questions

**Comments to the Authors:**

Reviewer #1: I thank the authors for addressing my previous comments. I am happy with the revision.

Reviewer #2: "Thus, if let the NB distribution have the same value...": The equation at the end of this sentence is very confusing and perhaps has a typo. k_D should be k_D hat? And equations need to be separated by commas; otherwise they look like a product.

Once this is corrected I believe the manuscript is ready to be published.

**Have the authors made all data and (if applicable) computational code underlying the findings in their manuscript fully available?**

Reviewer #1: None

Reviewer #2: Yes

PLOS authors have the option to publish the peer review history of their article (what does this mean?). If published, this will include your full peer review and any attached files.

Reviewer #1: No

Reviewer #2: No

---

## [Editor Report · Acceptance letter]

22 Jun 2022

PCOMPBIOL-D-21-01784R1 

Characterizing superspreading potential of infectious disease: Decomposition of individual transmissibility

Dear Dr Zhao,

I am pleased to inform you that your manuscript has been formally accepted for publication in PLOS Computational Biology. Your manuscript is now with our production department and you will be notified of the publication date in due course.

With kind regards,

Zsofi Zombor
